# GPU-HADVPPM4HIP V1.0: using the heterogeneous interface for portability (HIP) to speed up the piecewise parabolic method in the CAMx (v6.10) air quality model on China's domestic GPU-like accelerator

**Kai Cao[1], Qizhong Wu[1,5], Lingling Wang[2], Hengliang Guo[3], Nan Wang[2], Huaqiong Cheng[1,5], Xiao Tang[4], Dongxing Li[1,5], Lina Liu[3], Dongqing Li[1], Hao Wu[3], and Lanning Wang[1,5]**

[1]College of Global Change and Earth System Science, Faculty of Geographical Science, Beijing Normal University, Beijing 100875, China

[2]Henan Ecological Environmental Monitoring Centre and Safety Center, Henan Key Laboratory of Environmental Monitoring Technology, Zhengzhou 450008, China

[3]National Supercomputing Center in Zhengzhou, Zhengzhou, 450001, China

[4]State Key Laboratory of Atmospheric Boundary Layer Physics and Atmospheric Chemistry, Institute of Atmospheric Physics, Chinese Academy of Science, Beijing 100029, China

[5]Joint Center for Earth System Modeling and High Performance Computing, Beijing Normal University, Beijing, 100875, China

**Correspondence to:** Qizhong Wu (wqizhong@bnu.edu.cn); Lingling Wang(928216422@qq.com); Lanning Wang (wangln@bnu.edu.cn)

**Abstract.** Graphics processing units (GPUs) are becoming a compelling acceleration strategy for geoscience numerical model due to their powerful computing performance. In this study, AMD's heterogeneous compute interface for portability (HIP) was implemented to port the GPU acceleration version of the Piecewise Parabolic Method (PPM) solver (GPU-HADVPPM) from the NVIDIA GPUs to China's domestically GPU-like accelerators as GPU-HADVPPM4HIP. Further, it introduced the multi-level hybrid parallelism scheme to improve the total computational performance of the HIP version of the CAMx (CAMx-HIP) model on China's domestically heterogeneous cluster. The experimental results show that the acceleration effect of GPU-HADVPPM on the different GPU accelerators is more apparent when the computing scale is more

extensive, and the maximum speedup of GPU-HADVPPM on the domestic GPU-like accelerator is 28.9 times. The hybrid parallelism with a message passing interface (MPI) and HIP enables achieving up to 17.2 times speedup when configuring 32 CPU cores and GPU-like accelerators on the domestic heterogeneous cluster. The OpenMP technology is introduced further to reduce the computation time of the CAMx-HIP model by 1.9 times. More importantly, by comparing the simulation results of GPU-HADVPPM on NVIDIA GPUs and domestic GPU-like accelerators, it is found that the simulation results of GPU-HADVPPM on domestic GPU-like accelerators have less difference than the NVIDIA GPUs. Furthermore, we also exhibit that the data transfer efficiency between CPU and GPU has a meaningful essential impact on heterogeneous computing and point out that optimizing the data transfer efficiency between CPU and GPU is one of the critical directions to improve the computing efficiency of geoscience numerical models in heterogeneous clusters in the future.

## 1. Introduction

Over recent years, GPUs have become a necessary part of providing processing power for high-performance computing (HPC) applications, and heterogeneous supercomputing based on CPU processors and GPU accelerators has become the trend of global advanced supercomputing development. The 61st edition of the top 10 list, released in June 2023, reveals that 80% of advanced supercomputers adopt heterogeneous architectures (Top500, 2023). The Frontier system equipped with AMD Instinct MI250X GPU at the Oak Ridge National Laboratory remains the only actual exascale machine with the High-Performance Linpack benchmark (HPL) score of 1.194 Exaflop/s (News, 2023). How to realize large-scale parallel computing and improve the computational performance of geoscience numerical models on the GPU has become one of the significant directions for the future development of numerical models.

Regarding the heterogeneous porting for air quality model, most scholars select the chemical module, one of the hotspots, to implement heterogeneous porting, and porting the computational process initially on the CPU processes to the GPU accelerator, to improve the computing efficiency. For example, Sun et al. (2018) used CUDA technology to port the second-order Rosenbrock solver of the chemistry module of CAM4-Chem to NVIDIA Tesla K20X GPU. They

achieved up to 11.7x speedup compared to the AMD Opteron™ 6274 CPU (16 cores) using one
CPU core. Alvanos and Christoudias (2017) developed software that automatically generates
CUDA kernels to solve chemical kinetics equations in the chemistry module for the global climate
model ECHAM/MESSy Atmospheric Chemistry (EMAC), and performance evaluation shows a
20.4x speedup for the kernel execution. Linford et al. (2011) presented the Kinetic PreProcessor
(KPP) to generate the chemical mechanism code in CUDA language, which can be implemented
on the NVIDIA Tesla C1060 GPU. The KPP-generated SAPRC'99 mechanism from the CMAQ
model achieved a maximum speedup of 13.7x, and the KPP-generated RADM2 mechanism from
the WRF-chem model achieved an 8.5x speedup both compared to the Intel Quad-Core Xeon
5400 series CPU. Similarly, the advection module is also one of the hotspot modules in the air
quality model. Cao et al. (2023) adopted the Fortran-C-CUDA C scheme and implemented a series
of optimizations, including reducing the CPU–GPU communication frequency, optimizing the
GPU memory access, and thread and block co-indexing, to increase the computational efficiency
of the HADVPPM advection solver. It can achieve up to the 18.8x speedup on the NVIDIA Tesla
V100 GPU compared to the Intel Xeon Platinum 8168 CPU.
The CUDA technology was implemented to carry out heterogeneous porting for the
atmospheric chemical models from the CPU processors to different NVIDIA GPU accelerators. In
this study, the Heterogeneous-computing Interface for Portability (HIP) interface was introduced
to implement the porting of GPU-HADVPPM from the NVIDIA GPU to China's domestically
GPU-like accelerators based on the research of Cao et al. (2023). The domestic GPU-like
accelerator plays the same role as the NVIDIA GPU, which is also used to accelerate the
advection module in the CAMx model, so we refer to it as a GPU-like accelerator. First, we
compared the simulation results of the Fortran version CAMx model with the CAMx-CUDA and
CAMx‑HIP models, which were coupled with the CUDA and HIP versions of the GPU-
HADVPPM program, respectively. Then, the computing performance of GPU-HADVPPM
programs on different GPUs were compared. Finally, we tested the total performance of the
CAMx-HIP model with multi-level hybrid parallelization on China 's domestically heterogeneous
cluster.

## 2.  Model and experimental platform

### 2.1.  The CAMx model description and configuration

The Comprehensive Air Quality Model with Extensions version 6.10 (CAMx v6.10; ENVIRON, 2014) is a state-of-the-art air quality model that simulates the emission, dispersion, chemical reaction, and removal of the air pollutants on a system of nested three-dimensional grid boxes (CAMx, 2023). The Eulerian continuity equation is expressed as shown by Cao et al. (2023): the first term on the right-hand side represents horizontal advection, the second term represents net resolved vertical transport across an arbitrary space and time-varying height grid, and the third term represents turbulent diffusion on the sub-grid scale. Pollutant emission represents both point source emissions and grided source emissions. Chemistry is treated by solving a set of reaction equations defined by specific chemical mechanisms. Pollutant removal includes both dry deposition and wet scavenging by precipitation.

In terms of the horizontal advection term on the right-hand side, this equation is solved using either the Bott (1989) scheme or the Piecewise Parabolic Method (PPM) (Colella and Woodward, 1984; Odman and Ingram, 1996) scheme. The PPM horizontal advection scheme (HADVPPM) was selected in this study because it provides higher accuracy with minimal numerical diffusion (ENVIRON, 2014). The other numerical schemes selected during the CAMx model testing are listed in Table S1. As described by Cao et al. (2023), the -fp-model precise compile flag which can force the compiler to use the vectorization of some computation under value safety, is 41.4% faster than the -mieee-fp compile flag, which comes from the Makefile of the official CAMx version with the absolute errors of the simulation results are less than $\pm0.05$ ppbV. Therefore, the -fp-model precise compile flag was selected when compiling the CAMx model in this research.

### 2.2.  CUDA and ROCm introduction

Compute Unified Device Architecture (CUDA; NVIDIA, 2020) is a parallel programming paradigm released in 2007 by NVIDIA. CUDA is a proprietary application programming interface (API) and is only supported on NVIDIA's GPUs. CUDA programming uses a programming language similar to standard C, which achieves efficient parallel computing of programs on

NVIDIA GPUs by adding some keywords. The previous study implemented CUDA technology to port the HADVPPM program from CPU to NVIDIA GPU (Cao et al., 2023).

Radeon Open Compute platform (ROCm; AMD, 2023) is an open-source software platform developed by AMD for HPC and hyperscale GPU computing. The ROCm for the AMD GPU is generally equivalent to CUDA for NVIDIA GPU. The ROCm software platform uses the AMD's HIP interface, a C++ runtime API allowing developers to run programs on AMD GPUs. In general, they are very similar, and their code can be converted directly by replacing the string "cuda" with "hip" in most cases. More information about HIP API is available on the AMD ROCm website (ROCm, 2023). Similar to AMD GPU, developers can also use the ROCm-HIP programming interface to implement programs running on China's domestically GPU-like accelerator. The CUDA code cannot run directly on domestic GPU-like accelerators and must be transcoded into HIP code.

## 2.3. Hardware components and software environment of the testing system

Table 1 lists four GPU clusters where we conducted the experiments, two NVIDIA heterogeneous clusters that have the same hardware configuration as Cao et al. (2023), and two China's domestically heterogeneous clusters newly used in this research, namely "Songshan" supercomputer and "Taiyuan" computing platform. Two NVIDIA heterogeneous clusters are equipped with NVIDIA Tesla K40m and V100 GPU accelerators. Both domestic clusters include thousands of computing nodes, each contains one China's domestically CPU processor, four China's domestically GPU-like accelerators, and 128 GB of DDR4 2666 memory. The domestic CPU has four NUMA nodes, and each NUMA node has eight X86-based processors. The accelerator adopts a GPU-like architecture consisting of a 16 GB HBM2 device memory, and many compute units. The GPU-like accelerators are connected to the CPU with PCI-E, and the peak bandwidth of the data transfer between main memory and device memory is 16 GB/s.

It is worth noting that the "Taiyuan" computing platform has been updated in three main aspects compared to the "Songshan" supercomputer. The CPU clock speed has been increased from 2.0 GHz to 2.5 GHz, the number of GPU-like computing units has been increased from 3,840 to 8,192, and the peak bandwidth between main memory and video memory has been

increased from 16 GB/s to 32 GB/s. Regarding the software environment, the NVIDA GPU is
programmed using the CUDA toolkit, and the domestic GPU-like is programmed using the
ROCm-HIP toolkit developed by AMD (ROCm, 2023). More details about the hardware
composition and software environment of the four heterogeneous clusters are presented in Table 1.
**Table 1.** Configurations of the NVIDIA K40m cluster, NVIDIA V100 cluster, "Songshan" supercomputer, and
"Taiyuan" computing platform.

| | Hardware components | |
| --- | --- | --- |
| | CPU | GPU |
| NVIDIA K40m cluster | Intel Xeon E5-2682 v4 CPU @2.5 GHz, 16 cores | NVIDIA Tesla K40m GPU, 2880 CUDA cores, 12 GB video memory |
| NVIDIA V100 cluster | Intel Xeon Platinum 8168 CPU @2.7 GHz, 24 cores | NVIDIA Tesla V100 GPU, 5120 CUDA cores, 16 GB video memory |
| Songshan supercomputer | China's domestically CPU processor A, 2.0GHz, 32 cores | China's domestically GPU-like accelerator A, 3840 computing units, 16 GB memory |
| Taiyuan computing platform | China's domestically CPU processor B, 2.5GHz, 32 cores | China's domestically GPU-like accelerator B, 8192 computing units, 16 GB memory |
| | Software environment | |
| | Compiler and MPI | Programming model |
| NVIDIA K40m cluster | Intel Toolkit 2021.4.0 | CUDA-10.2 |
| NVIDIA V100 cluster | Intel Toolkit 2019.1.144 | CUDA-10.0 |
| Songshan supercomputer | Intel Toolkit 2021.3.0 | ROCm-4.0.1/ DTK-23.04 |
| Taiyuan computing platform | Intel Toolkit 2021.3.0 | DTK-23.04 |

## 3. Implementation details

This section mainly introduced the strategy of porting the HADVPPM program from CPU to
NVIDIA GPU and domestic GPU-like accelerator, as well as the proposed multi-level hybrid
parallelism technology to make full use of computing resources.

## 3.1. Porting the HADVPPM program from CPU to NVIDIA GPU and domestic GPU-like accelerator

Fig.1 shows the heterogeneous porting process of HADVPPM from CPU to NVIDIA GPU
and domestic GPU-like accelerator. First, the original Fortran code was refactored using standard
C language. Then, the CUDA and ROCm HIP technology were used to convert the standard C
code into CUDA C and HIP C code to make it computable on the NIVIDA GPU and domestic
GPU-like accelerator. Similar to CUDA technology, HIP technology is implemented to convert the
standard C code to HIP C code by adding related built-in functions (such as hipMalloc,
hipMemcpy, hipFree, etc.). To facilitate the portability of applications across different GPU
platforms, ROCm provides Hipify toolkits to help transcode. The Hipify toolkit is essentially a
simple script written in the Perl language, and its function is text replacement, which replaces the
function name in CUDA C code with the corresponding name in HIP C code according to specific
rules. For example, the Hipify toolkit can automatically recognize and replace the memory
allocation function cudaMalloc in CUDA with hipMalloc. Therefore, the thread and block
configuration of the GPU remains unchanged due to the simple text substitution during the
transcoding. In this study, the ROCm HIP technology was used to implement the operation of
GPU-HADVPPM on the domestic GPU-like accelerator based on the CUDA version of GPU-
HADVPPM developed by Cao et al. (2023). The HIP code was compiled using the "hipcc"
compiler driver with the library flag "-lamdhip64".

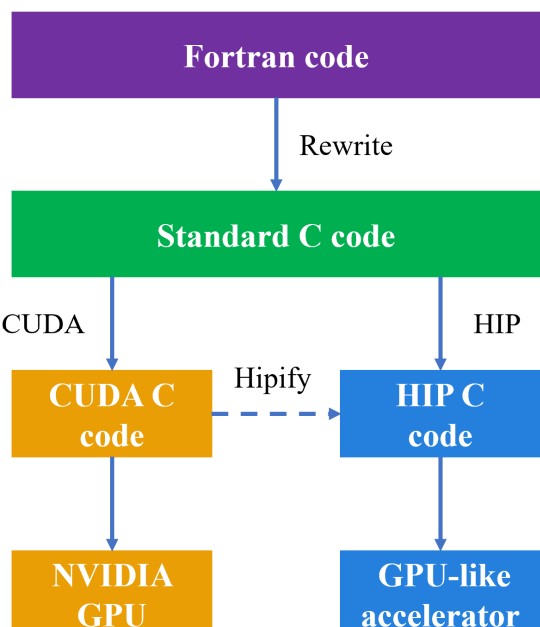


**Figure 1.** The heterogeneous porting process of HADVPPM Fortran code from CPU to NVIDIA GPU and
domestic GPU-like accelerator.

## 3.2. Multi-level hybrid parallelization of CAMx model on heterogeneous platform

The original CAMx model running on the CPUs supports two types of parallelization (ENVIRON, 2014): (1) OpenMP (OMP), which supports multi-platform (e.g., multi-core) shared-memory programming in C/C++ and Fortran; (2) Message Passing Interface (MPI), which is a message passing interface standard for developing and running parallel applications on the distributed-memory computer cluster. During the process of CAMx model simulation, MPI and OMP hybrid parallelism can be used, several CPU processes can be launched, and each process can spawn several threads. This hybrid parallelism can significantly improve the computational efficiency of the CAMx model.

As mentioned, the original CAMx model supports message passing interface (MPI) parallel technology running on the general-purpose CPU. The simulation domain is divided into several sub-regions by MPI, and each CPU process is responsible for the computation of its sub-region. To expand the heterogeneous parallel scale of the CAMx model on the Songshan supercomputer, a hybrid parallel architecture with an MPI and HIP was adopted to make full use of GPU computing resources. Firstly, we use the ROCm-HIP library function hipGetDeviceCount to obtain the number of GPU accelerators configured for each compute node. Then, the total number of accelerators to be launched and the ID number of accelerator cards in each node were determined according to the MPI process ID number and the remainder function in standard C language. Finally, the hipSetDevice library function in ROCm-HIP is used to configure an accelerator for each CPU core.

This study uses GPU-HADVPPM with an MPI and HIP heterogeneous hybrid programming technology to run on multiple domestic GPU-like accelerators. However, the number of GPU-like accelerators in a single compute node is usually much smaller than the number of CPU cores in heterogeneous HPC systems. Therefore, to make full use of the remaining CPU computing resources, the OMP API of the CAMx model is further introduced to realize the MPI+OMP hybrid parallelism of other modules on the CPU. A schematic of the multi-level hybrid parallel framework is shown in Fig.2. For example, four CPU processes and four GPU-like accelerators are launched in a computing node, and each CPU process spawns four threads. Then the advection

module is simulated by 4 GPU-like accelerators, and 4*4 threads spawned by CPU processes do
the other modules.

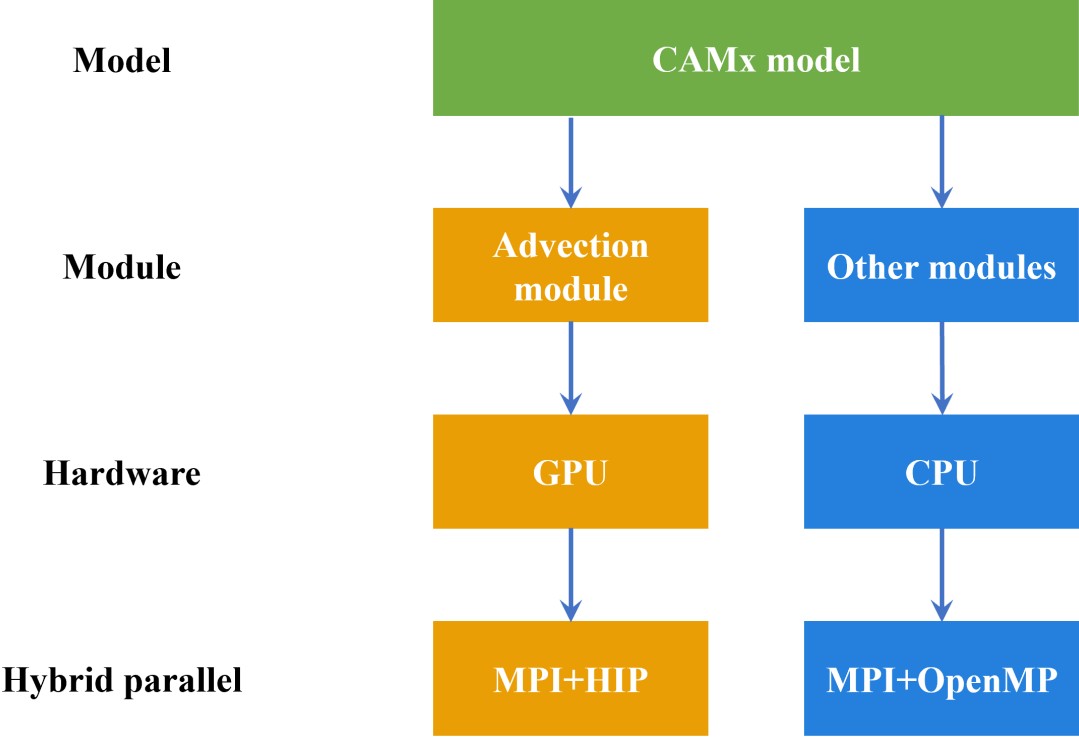

**Figure 2.** A schematic of the multi-level hybrid parallel framework.

## 4. Results and evaluation

The computational performance experiments of CUDA and HIP version GPU-HADVPPM
are reported in this section. First, we compared the simulation result of the Fortran version CAMx
model with the CAMx-CUDA and CAMx-HIP models, which were coupled with the CUDA and
HIP versions of the GPU-HADVPPM program, respectively. Then, the computational
performance of GPU-HADVPPM programs on the NVIDIA GPU and domestic GPU-like
accelerator are compared. Finally, we tested the total performance of the CAMx-HIP model with
multi-level hybrid parallelization on the "Songshan" supercomputer. For ease of description, the
CAMx versions of the HADVPPM program written in Fortran, CUDA C, and HIP C code are
named Fortran, CUDA, and HIP, respectively.

## 4.1. Experimental setup

Three test cases were used to evaluate the performance of CUDA and HIP version GPU-HADVPPM. The experimental setup for the three test cases is shown in Table 2. In the previous study of Cao et al. (2023), the BJ case was used to carry out the performance tests, and the HN case and ZY case was the newly constructed test cases in this study. The Beijing case (BJ) covers Beijing, Tianjin, and part of the Hebei Province with $145 \times 157$ grid boxes, and the simulation of the BJ case starts on 1 November, 2020. The Henan case (HN) mainly covers the Henan Province with $209 \times 209$ grid boxes. The starting date of simulation in the HN case is 1 October, 2022. The Zhongyuan case (ZY) has the widest coverage of the three cases, with Henan Province as the center, covering the Beijing-Tianjin-Hebei region, Shanxi Province, Shaanxi Province, Hubei Province, Anhui Province, Jiangsu Province, and Shandong Province, with $531 \times 513$ grid boxes. ZY case started simulation on 4 January, 2023. All three performance test cases have a 3km horizontal resolution, 48 hours of simulation, and 14 vertical model layers. The number of three-dimensional grid boxes in BJ, HN, and ZY cases total 318,710, 611,534, and 3,813,642, respectively. The meteorological fields inputting the different versions of the CAMx model in the three cases were provided by the Weather Research and Forecasting Model (WRF). In terms of emission inventories, the emission for the BJ case is consistent with the Cao et al. (2023), the HN case uses the Multi-resolution Emission Inventory for China (MEIC). The ZY case uses the emission constructed by the Sparse Matrix Operator Kernel Emission (SMOKE) model in this study.

**Table 2.** The experimental setup for the BJ, HN, and ZY cases.

|  | **BJ** | **HN** | **ZY** |
|---|---|---|---|
| **Start date** | November 1, 2020 | October 1, 2022 | 1 January, 2023 |
| **Horizontal resolution** | 3km | 3km | 3km |
| **Grid boxes** | $145 \times 157 \times 14$ | $209 \times 209 \times 14$ | $531 \times 513 \times 14$ |
| **Meteorological fields** | WRF | WRF | WRF |
| **Emission** | Cao et al. (2023) | MEIC | SMOKE |

## 4.2. Error analysis

The hourly concentrations of four major species, i.e., $O_3$, $PSO_4$, $CO$, and $NO_2$, outputted by the Fortran, CUDA, and HIP versions of CAMx for the BJ case are compared to verify the results

correctness before testing the computational performance. Fig.3 shows the four major species
simulation results of the three CAMx versions, including the Fortran version on the Intel E5-2682
v4 CPU, the CUDA version on the NVIDIA K40m cluster, and the HIP version on the "Songshan"
supercomputer, after 48 hours integration, as well as the absolute errors (AEs) of their
concentrations. As described by Cao et al. (2023), the parallel design of the CAMx model adopts
the primary/secondary mode, and the P0 process is responsible for inputting and outputting the
data and calling the MPI_Barrier function to synchronize the process, and the other processes are
accountable for simulation. When comparing the simulation results, we only launched 2 CPU
processes on the CPU platform, launched 2 CPU processes and configured 2 GPU accelerators on
the NVIDIA K40m cluster and "Songshan" supercomputer, respectively.
The species' spatial pattern of the three CAMx versions on different platforms are visually
very consistent. The AEs between the HIP and Fortran versions are much smaller than the CUDA
and Fortran versions. For example, the AEs between the CUDA and Fortran versions for $O_3$, $PSO_4$,
and $NO_2$ are in the range of $\pm0.04$ ppbV, $\pm0.02$ $\mu g \cdot m^{-3}$, and $\pm0.04$ ppbV. The AEs between the
HIP and Fortran versions for the three species fall into the range of $\pm0.01$ ppbV, $\pm0.005$ $\mu g \cdot m^{-3}$,
and $\pm0.01$ ppbV. For CO, AEs are relatively large due to their high background concentration.
However, the AEs between the HIP and Fortran versions are also less than those between the
CUDA and Fortran versions, which were in the range of $\pm0.4$ ppbV and $\pm0.1$ ppbV, respectively.
Considering the situation of AEs accumulation and growth, Fig.4 highlights the time series of
AEs between Fortran and CUDA versions and between Fortran and HIP versions after grid
averaging. As is shown in Fig.4, the AEs of $O_3$, $PSO_4$, CO, and $NO_2$ between the Fortran version
and the CUDA version are -0.0002 to 0.0001 ppbV, -0.00003 to 0.00001 $\mu g \cdot m^{-3}$, -0.0004 to
0.0004 ppbV, and -0.0002 to 0.0002 ppbV, respectively, and fluctuate. Although the AEs of the
above four species between the Fortran and the HIP version also fluctuate, the fluctuation range is
much smaller than that of the CUDA version. Notably, the AEs between Fortran and CUDA
versions and between Fortran and HIP versions do not accumulate and grow over prolonged
simulation periods.

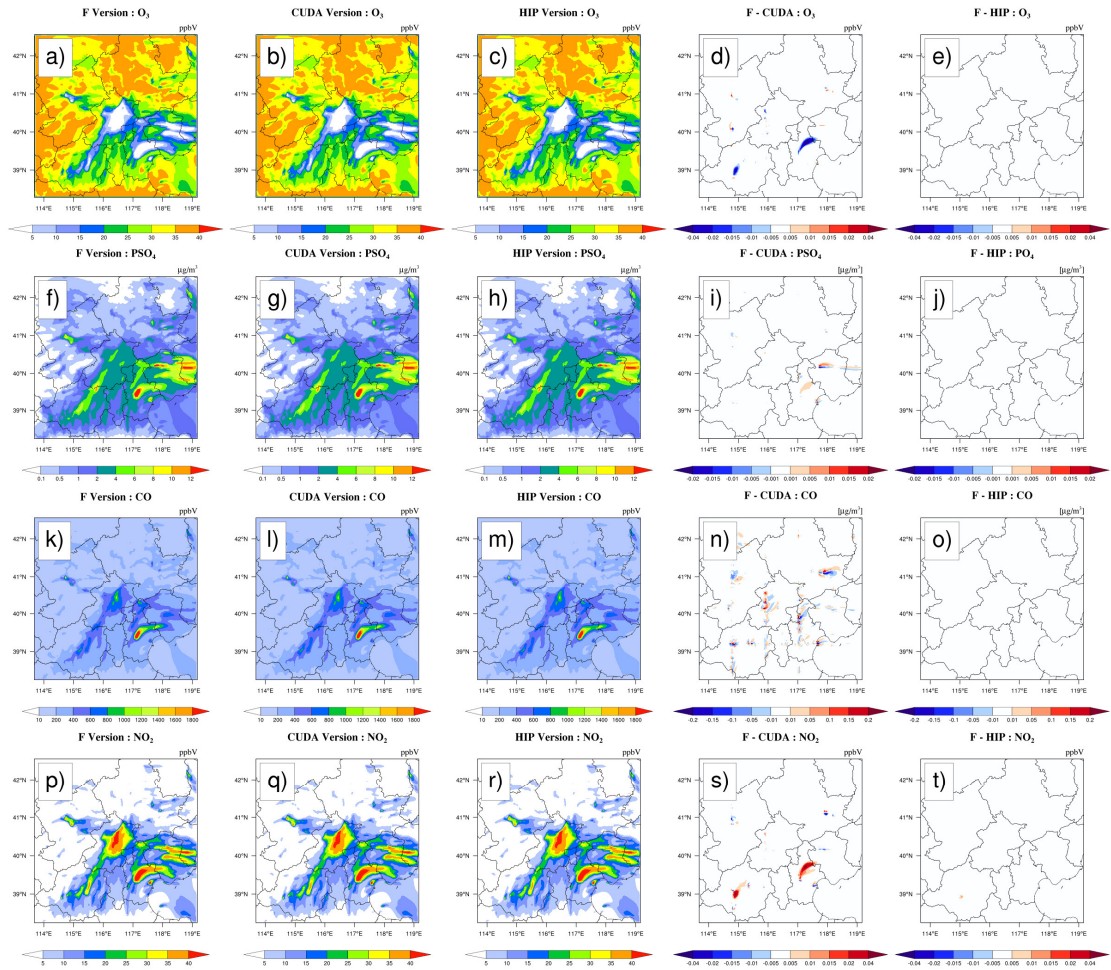


**Figure 3.** $O_3$, $PSO_4$, CO, and $NO_2$ concentrations outputted by the CAMx Fortran version on the Intel E5-2682 v4 CPU, CUDA version on the NVIDIA K40m cluster, and HIP version on the "Songshan" supercomputer under the BJ case. Panels (a), (f), (k), and (p) are from the Fortran version of simulation results for four species. Panels (b), (g), (l), and (q) are from the CUDA version of simulation results for four species. Panels (c), (h), (m), and (r) are from the HIP version of simulation results for four species. Panels (d), (i), (n), and (s) are the AEs between the Fortran and CUDA versions. Panels (e), (j), (o), and (t) are the AEs between the Fortran and HIP versions.

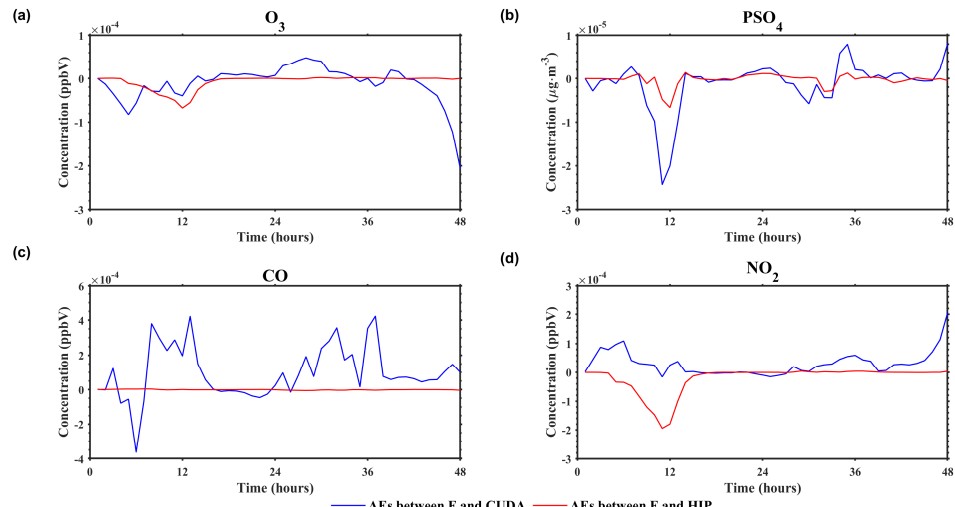

**Figure 4.** After grid averaging, the time series of AEs between Fortran and CUDA versions (solid blue line) and between Fortran and HIP versions (solid red line). Panel (a)~(d) represent the AEs of $O_3$, $PSO_4$, CO, and $NO_2$, respectively.

To further detail the differences in the simulation results, we supplement the offline experimental results of the advection module on the NVIDIA K40m cluster and the Songshan supercomputer. First, we construct the Fortran programs to provide consistent input data for the advection module written in CUDA C code and HIP C code on NVIDIA Tesla K40m GPU and domestic GPU-like accelerator, respectively. The accuracy of the input data is kept at 12 decimal places. Then, the advection module outputs and prints the computing results after completing one integration operation on different accelerators. Finally, the results of the various accelerators were compared with those of the Fortran code on the Intel Xeon E5-2682 v4 CPU processor. The specific results are shown in the Fig.5. The difference in the computing results of the advection module written in HIP C code on the domestic GPU-like accelerator is smaller than that of the CUDA C code on the NVIDIA Tesla K40m GPU. The mean relative errors (REs) and AEs of the computing results on the NVIDIA Tesla K40m GPU are $1.3 \times 10^{-5}$ % and $7.1 \times 10^{-9}$, respectively, while on the domestic GPU-like accelerator, the mean REs and AEs of the results are $5.4 \times 10^{-6}$% and $2.6 \times 10^{-9}$, respectively.

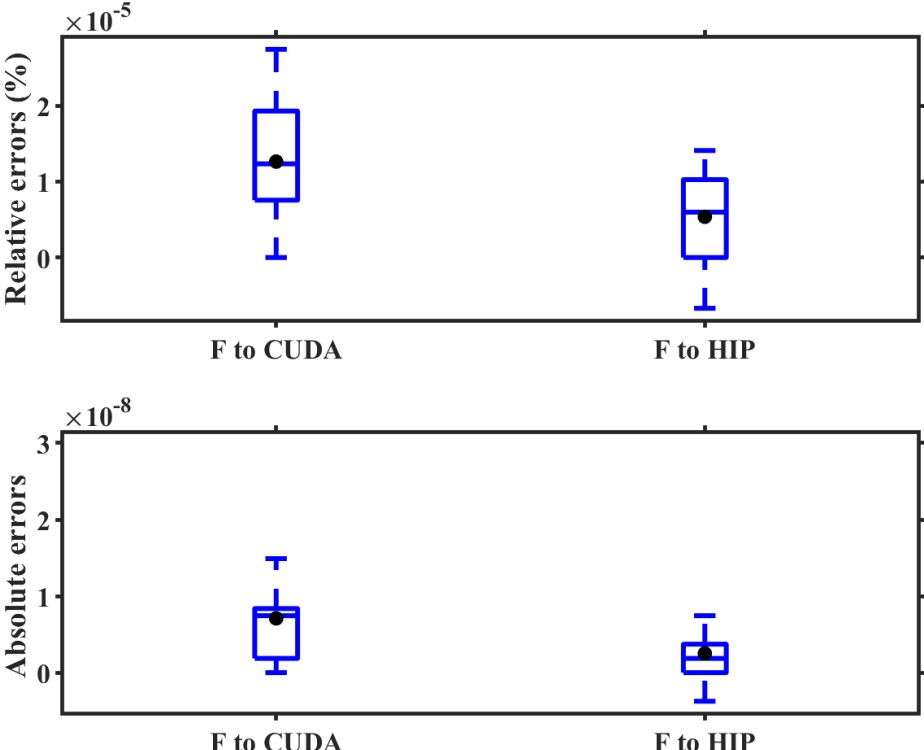

**Figure 5.** The boxplots of REs and AEs between the Fortran code on Intel Xeon E5-2682 v4 CPU and CUDA C code on NVIDIA Tesla K40m GPU, and between HIP C code on domestic GPU-like accelerator, respectively, in the case of offline testing.

Fig.6 further presents the boxplot of the REs in all grid boxes for the $PSO_4$, $PNO_3$, $PNH_4$, $O_3$, CO, and $NO_2$ during the 48-hour simulation under the BJ case. Statistically, the REs between the CUDA version on the NVIDIA K40m cluster and Fortran version on the Intel E5-2682 v4 CPU for the above six species are in the range of $\pm0.006\%$, $\pm0.01\%$, $\pm0.008\%$, $\pm0.002\%$, $\pm0.002\%$, and $\pm0.002\%$. In terms of REs between the HIP version on the "Songshan" supercomputer and the Fortran version on the Intel E5-2682 v4 CPU, the values are much smaller than REs between CUDA and Fortran versions which fall into the range of $\pm0.0005\%$, $\pm0.004\%$, $\pm0.004\%$, $\pm0.00006\%$, $\pm0.00004\%$, and $\pm0.00008\%$, respectively. In the air quality model, the initial concentration of secondary fine particulate matter such as $PSO_4$, $PNO_3$, and $PNH_4$ is very low and is mainly generated by complex chemical reactions. The integration process of the advection module is ported from the CPU processor to the GPU accelerator, which will lead to minor differences in the results due to different hardware. The low initial concentration of secondary fine particulate matter is sensitive to these minor differences, which may eventually lead to a higher

difference in the simulation results of secondary particulate matter than other species.

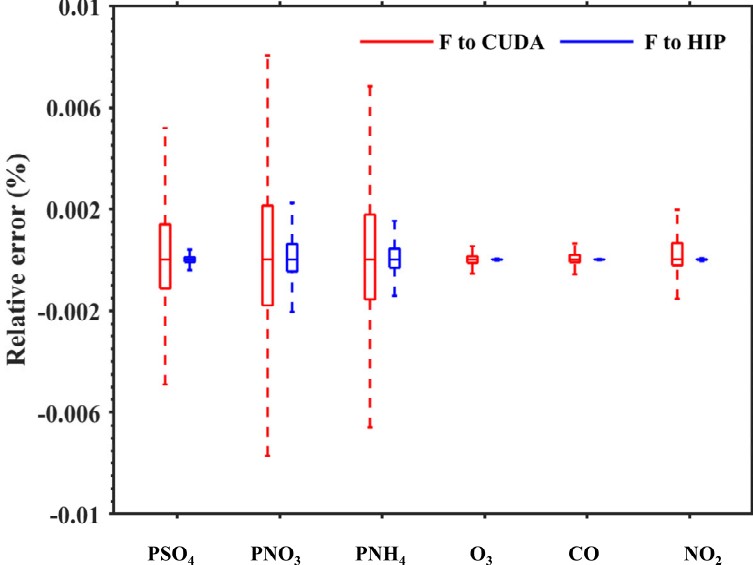


**Figure 6.** The REs distribution in all grid boxes for the $PSO_4$, $PNO_3$, $PNH_4$, $O_3$, CO, and $NO_2$ under the BJ case.
The red boxplot represents the REs between the CUDA version on the NVIDIA K40m cluster and the Fortran
version on the Intel E5-2682 v4 CPU, and the blue boxplot represents the REs between the HIP version on the
"Songshan" supercomputer and the Fortran version on the Intel E5-2682 v4 CPU.

314       Wang et al. (2021) verified the applicability of the numerical model in scientific research by

computing the ratio of root mean square error (RMSE) between two different model versions to
system spatial variation (standard deviation, std). If the ratio is smaller, it is indicated that the
difference in the simulation results of the model on the GPU is minimal compared with the spatial
variation of the system. That is to say, the simulation results of the model on the GPU are accepted
for scientific research. Here, we calculate the standard deviation of $O_3$, $PSO_4$, CO, and $NO_2$ on the
Intel Xeon E5-2682 v4 CPU and their RMSE between the NVIDIA V100 cluster, NVIDIA K40m
cluster, and "Songshan" supercomputer and the Intel Xeon E5-2682 v4 CPU, which are presented
in Table 3. The std for the above four species on the Intel Xeon E5-2682 v4 CPU are 9.6 ppbV, 1.7
$\mu g \cdot m^{-3}$, 141.9 ppbV, and 7.4 ppbV, respectively, and their ratios of RMSE and std on the
"Songshan" supercomputer are $5.8 \times 10^{-5}$ %, $4.8 \times 10^{-6}$ %, $5.7 \times 10^{-8}$ %, and $2.1 \times 10^{-4}$ %,
which are smaller than two NVIDIA clusters, significantly much smaller than the NVIDIA V100
cluster. For example, the ratio on the NVIDIA K40m cluster for four species are $1.2 \times 10^{-4}$ %,
$6.6 \times 10^{-5}$ %, $7.0 \times 10^{-5}$ %, and $4.1 \times 10^{-4}$ %, and ratio on the NVIDIA V100 cluster are
$1.5 \times 10^{-2}\%$, $2.5 \times 10^{-3}\%$, $6.4 \times 10^{-3}\%$, and $1.3 \times 10^{-3}\%$, respectively.
**Table 3.** The standard deviation (std) of O₃, PSO₄, CO, and NO₂ on the Intel Xeon E5-2682 v4 CPU, root mean
square error (RMSE), and its ratio on the NVIDIA V100 cluster, NVIDIA K40m cluster, and "Songshan"
supercomputer

| | | NIVIDA V100 cluster | | NIVIDA K40m cluster | | "Songshan" supercomputer | |
|---|---|---|---|---|---|---|---|
| | std | RMSE | RMSE/std | RMSE | RMSE/std | RMSE | RMSE/std |
| $O_3$ (ppbV) | 9.6 | $1.5 \times 10^{-3}$ | $1.5 \times 10^{-2}$ | $1.1 \times 10^{-5}$ | $1.2 \times 10^{-4}$ | $7.4 \times 10^{-6}$ | $7.7 \times 10^{-5}$ |
| $PSO_4$ ($\mu g \cdot m^{-3}$) | 1.7 | $4.3 \times 10^{-5}$ | $2.5 \times 10^{-3}$ | $1.1 \times 10^{-6}$ | $6.6 \times 10^{-5}$ | $2.5 \times 10^{-7}$ | $1.5 \times 10^{-5}$ |
| $CO$ (ppbV) | 141.9 | $9.0 \times 10^{-3}$ | $6.4 \times 10^{-3}$ | $1.0 \times 10^{-4}$ | $7.0 \times 10^{-5}$ | $4.4 \times 10^{-7}$ | $3.1 \times 10^{-7}$ |
| $NO_2$ (ppbV) | 7.4 | $9.3 \times 10^{-5}$ | $1.3 \times 10^{-3}$ | $3.0 \times 10^{-5}$ | $4.1 \times 10^{-4}$ | $2.0 \times 10^{-5}$ | $2.7 \times 10^{-4}$ |

From AEs, REs, and the ratio of RMSE and std between different CAMx versions, there is
less difference that the GPU-HADVPPM4HIP program runs on the "Songshan" supercomputer.
Because the simulation accuracy of geoscience numerical model is closely related to the model
efficiency, and many model optimization works improve the computational performance by
reducing the precision of the data, such as Váňa et al. (2017) changed some variables precision in
the atmospheric model from double precision to single precision, which increased the overall
computational efficiency by 40%, and Wang et al. (2019) improved the computational efficiency
of the gas-phase chemistry module in the air quality mode by 25%~28% by modifying the
floating-point precision compile flag. Therefore, we speculate that this may be related to the
manufacturing process of NVIDIA GPUs and domestic GPU-like accelerators, which may use
unknown optimizations to improve GPU performance efficiency by losing part of the accuracy. In
this study, we mainly focus on numerical simulation. Of course, we also want to know the specific
reasons for this. Still, we are not professional GPU research and development designers after all
and do not know the underlying design logic of the hardware, so we can only present our
experimental results in the air pollution model to you, and discuss with each other to jointly
promote the application of GPU in the field of geoscience numerical models.
**4.3. Application performance**
**4.3.1. GPU-HADVPPM on a single GPU accelerator**
As described in Sect. 4.2, we validate the 48-hour simulation results outputted by the Fortran,
CUDA, and HIP versions of CAMx model. Next, computational performance was compared for
the Fortran version of HADVPPM on the Intel Xeon E5-2682 v4 CPU and domestic CPU
processor A, the CUDA version of GPU-HADVPPM on the NVIDIA Tesla K40m and V100 GPU,
and the HIP version of GPU-HADVPPM on the domestic GPU-like accelerator A, under the BJ,
HN and ZY case. The simulation time in this section is 1 hour unless otherwise specified.
Similarly, since the CAMx model adopts the primary/secondary mode, two CPU processes,
P0 and P1, are launched on the CPU, and the system_clock functions in the Fortran language are
used to test the elapsed time of the advection module in the P1 process. When testing the
computation performance of the advection module on the GPU-like accelerator, we only launch 2
CPU processes and 2 GPU-like accelerators. When a P1 process runs to the advection module, the
original computation process is migrated from the CPU to the GPU, and the hipEvent_t function
in HIP programming is used to test the running time of the advection module on the GPU-like
accelerator. When comparing the speedup on different GPU accelerators, the elapsed time of the
advection module launched one CPU process (P1) on the domestic CPU processor A is taken as
the benchmark; that is, the speedup is 1.0x. The runtime of the advection module on Intel CPU
processor and different GPU accelerators is compared with the baseline to obtain the speedup.
Fig.7 (a) and (b) show the elapsed time and speedup of the different versions of HADVPPM
on the CPU processors and GPU accelerators for BJ, HN, and ZY cases, respectively. The results
show that CUDA and HIP technology to port HADVPPM from CPU to GPU can significantly
improve its computational efficiency. For example, the elapsed time of the advection module on
the domestic processor A is 609.2 seconds under the ZY case. After it is ported to the domestic
GPU accelerator and NVIDIA Tesla V100 GPU, it only takes 21.1 seconds and 7.6 seconds to
complete the computing, and the speedups are 28.9x and 80.2x, respectively. The ZY case had the
most significant number of grids in the three cases. It exceeded the memory of a single NVIDIA
Tesla K40m GPU accelerator, so it was not possible to test its elapsed time on it. Moreover, the
optimization of thread and block co-indexing is used to compute the grid point in the horizontal
direction simultaneously (Cao et al., 2023). Therefore, it can be seen from Fig. 6(b) that the larger
the computing scale, the more pronounced the acceleration, which indicates that GPU is more
suitable for super-large scale parallel computing and provides technical support for accurate and
fast simulation of ultra-high-resolution air quality at the meter level in the future.

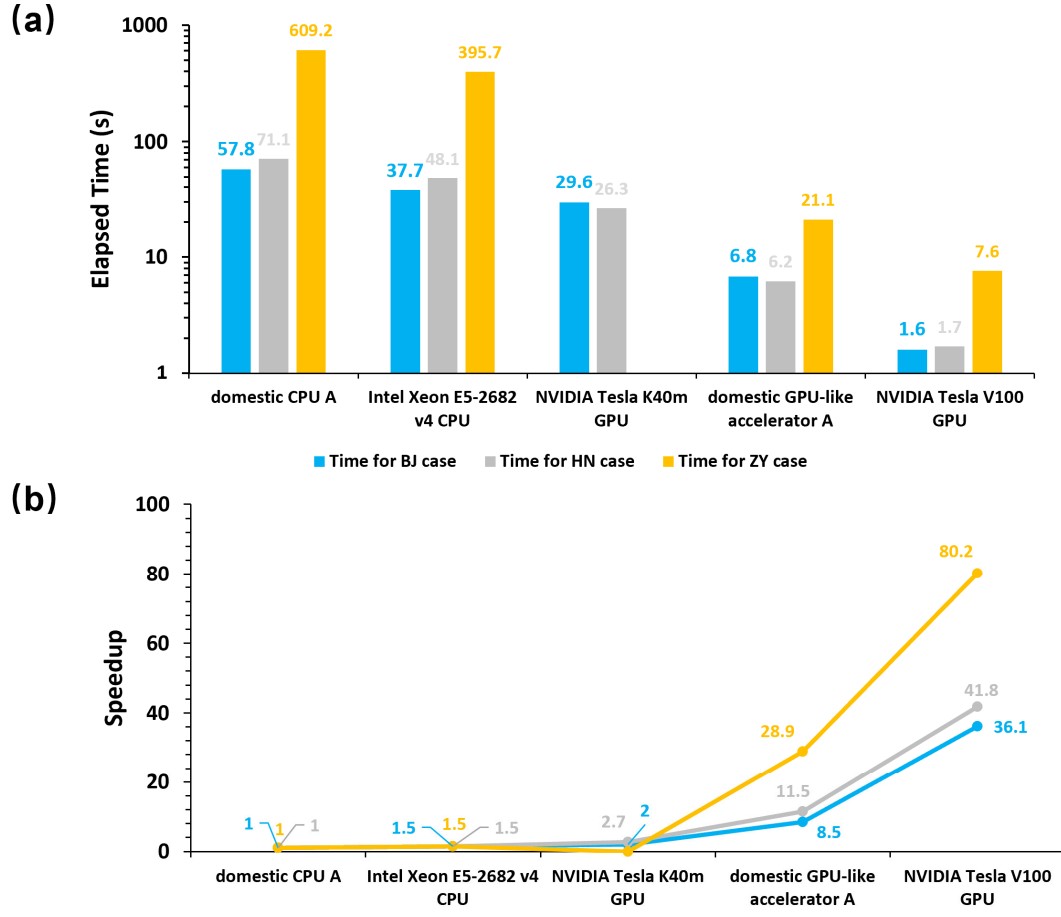


**Figure 7.** The elapsed time (a) and speedup (b) of the Fortran version of HADVPPM on the Intel Xeon E5-2682
v4 CPU and the domestic CPU processor A, the CUDA version of GPU-HADVPPM on the NVIDIA Tesla K40m
GPU, NVIDIA Tesla V100 GPU, and the HIP version of GPU-HADVPPM on the domestic GPU-like accelerator
A for BJ, HN, and ZY case. The unit of elapsed time is in seconds (s).

386        The BJ, HN, and ZY case timestep were 59, 47, and 61, respectively. Fig.8 shows the GPU-

HADVPPM4HIP acceleration in each time step on a single domestic GPU-like accelerator A. It
can be seen from the figure that all three cases have the smallest speedup of 8.2x, 11.2x, and 27.8x
at the first timestep, which is related to the time required for GPU-like accelerator startup. When
the GPU-like is started and operating normally, the speedup of the three cases tends to be stable in
the following time steps and stabilize around 8.5x, 11.5x, and 28.0x, respectively.

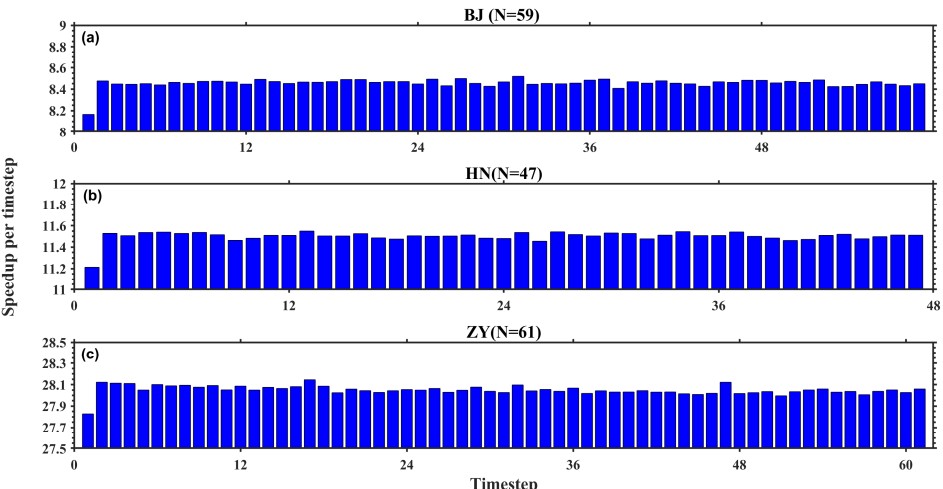

**Figure 8.** The GPU-HADVPPM4HIP acceleration in each time step on a single GPU-like accelerator for BJ, HN, and ZY cases. The timestep of the above three cases are 59, 47, and 61, respectively.

Table 4 further lists the total elapsed time of CAMx Fortran and HIP versions for BJ case on the "Songshan" supercomputer and "Taiyuan" computing platform and the computing time of the advection module with or without data transfer. By coupling the GPU-HADVPPM4HIP to the CAMx model and adopting a series of optimizations (Cao et al., 2023), such as communication optimization, memory access optimization, and 2D thread optimization, the overall computation time of the CAMx-HIP model on a single domestic GPU-like accelerator is faster than that of the original Fortran version on a single domestic CPU core. For example, on the "Songshan" supercomputer, one hour of simulation in the CAMx-HIP model takes 469 seconds, and the Fortran version takes 481 seconds. On the "Taiyuan" computing platform, the acceleration effect is more evident due to the upgrade of hardware and network bandwidth. The integration time of the CAMx-HIP model is 433 seconds when maintaining the same software environment, and the integration time of the Fortran version is 453 seconds.

The elapsed time of GPU-HADVPPM given in Table 4 on NVIDIA GPU and domestic GPU-like accelerator does not consider the data transfer time between CPU and GPU. However, the communication bandwidth of data transfer between the CPU and GPU is one of the most significant factors that restrict the performance of the numerical model on the heterogeneous cluster (Mielikainen et al., 2012; Mielikainen et al., 2013; Huang et al., 2013). To illustrate the significant impact of CPU-GPU data transfer efficiency, the computational performance of GPU-HADVPPM with and without data transfer time for the BJ case is tested on the "Songshan"

supercomputer and "Taiyuan" computing platform with the same DTK version 23.04 software
environment, and the results are further presented in Table 6. For convenience of description, we
refer to the execution time of GPU-HADVPPM program on GPU kernel as kernel execution time,
and the time of GPU-HADVPPM running on GPU as total runtime, which contains two parts,
namely, kernel execution time and data transfer time between CPU and GPU. After testing, the
kernel execution time and total running time of the GPU-HADVPPM4HIP program on domestic
GPU-like accelerator A are 6.8 and 29.8 seconds, respectively. In other words, it only takes 6.8
seconds to complete the computation on the domestic accelerator. Still, it takes 23.0 seconds to
complete the data transfer between the CPU and the domestic GPU-like accelerator, which is 3.4
times the computation time. The same problem exists in the more advanced the "Taiyuan"
computing platform, where the GPU-HADVPPM4HIP takes only 5.7 seconds to complete the
computation, while the data transmission takes 18.2 seconds, 3.2 times the computation time.

426        By comparing the kernel execution time and total running time of GPU-HADVPPM4HIP on

the domestic accelerator, it can be seen that the data transfer efficiency between CPU and GPU is
inefficient, which seriously restricts the computational performance of numerical models in
heterogeneous clusters. On the one hand, improving the data transfer bandwidth between CPU and
GPU can improve the computational efficiency of the model in heterogeneous clusters. On the
other hand, optimization measures can be implemented to improve the data transfer efficiency
between CPU and GPU. For example, (1) Asynchronous data transfer reduces the communication
latency between CPU and GPU. Computation and data transfer are performed simultaneously to
hide communication overhead; (2) Currently, some advanced GPU architectures support a unified
memory architecture, so that the CPU and GPU can share the same memory space and avoid
frequent data transfers. This reduces the overhead of data transfer and improves data transfer
efficiency; (3) Cao et al. (2023) adopted communication optimization measures to minimize the
communication frequency in one-time integration step to one, but there is still the problem of high
communication frequency in the whole simulation. In the future, we will consider porting other
hotspots of the CAMx model or even the entire integral module except I/O, to GPU-like
accelerators for increasing the proportion of code on the GPU and reducing the frequency of CPU-
GPU communication.

Video memory and bandwidth are the two most significant factors affecting GPU performance, and high video memory and high bandwidth can better play the powerful computing performance of GPUs. Usually, the memory and bandwidth of the GPU are already provided by the factory. In this case, the amount of data transferred to the GPU can be roughly estimated before the data is transferred to the GPU. The amount of data transferred to the GPU can be adjusted according to the size of the GPU memory to ensure that the amount of data transferred to the GPU each time reaches the maximum GPU video memory, to give full play to the GPU performance more efficiently.

**Table 4.** The total elapsed time of CAMx Fortran and HIP versions for the BJ case on the "Songshan" supercomputer and "Taiyuan" computing platform, and the computing time of the advection module with or without data transfer. The unit of elapsed time is in seconds (s).

| | "Songshan" supercomputer | | "Taiyuan" computing platform | |
|---|---|---|---|---|
| | Fortran version | HIP version | Fortran version | HIP version |
| Total elapsed time | 481.0 | 469.0 | 453.0 | 433.0 |
| Computing time of advection module without data transfer | 57.8 | 6.8 | 47.8 | 5.7 |
| Computing time of advection module with data transfer | 57.8 | 29.8 | 47.8 | 23.9 |

### 4.3.2. CAMx-HIP model on the heterogeneous cluster

Generally, heterogeneous HPC systems have thousands of compute nodes equipped with one or more GPUs on each compute node. To fully use multiple GPUs, the hybrid parallelism with an MPI and HIP paradigm was used to implement the HIP version of GPU-HADVPPM run on multiple domestic GPU-like accelerators. During the simulation of the CAMx model, the emission, advection, dry deposition, diffusion, wet deposition, photolysis process, and chemical process will be computed sequentially. In heterogeneous computing platforms, except for the advection process, the CPU processor completes the simulation of the rest of the processes, and the advection process is completed on the GPU accelerator. For example, using MPI and HIP hybrid parallel technology to launch four CPU processes and four GPU accelerators simultaneously, the advection process is completed on four GPUs, and the other processes are still completed on four CPU processes.

Fig.9 shows the total elapsed time and speedup of the CAMx-HIP model, which is coupled with the HIP version GPU-HADVPPM on the "Songshan" supercomputer under the BJ, HN, and

ZY cases. The simulation of the above three cases for one hour took 488 seconds, 1135 seconds,
and 5691 seconds, respectively, when launching two domestic CPU processors and two GPU-like
accelerators. For the BJ and HN case, the parallel scalability is highest when configured with 24
CPU cores and 24 GPU-like accelerators, with speedup of 8.1x and 11.6x, respectively. Regarding
the ZY case, due to its large number of grids, the parallel scalability is the highest when 32 CPU
cores and 32 GPU-like accelerators are configured, and the acceleration ratio is 17.2x.
As mentioned above, data transfer between CPU and GPU takes several times more time than
computation. Regardless of the CPU-GPU data transfer consumption, GPU-HADVPPM4HIP can
achieve up to 28.9x speedup on a single domestic GPU-like accelerator. However, in terms of the
total time consumption, the CAMx-HIP model is only 10~20 seconds faster than the original
Fortran version when one GPU-like accelerator is configured. As the number of CPU cores and
GPU-like accelerators increases, the overall computing performance of the CAMx-HIP model is
lower than that of the original Fortran version. The main reason is related to the amount of data
transferred to GPU. As the number of MPI processes increases, the number of grids responsible
for each process decreases, and the amount of data transmitted by the advection module from CPU
to GPU decreases. However, GPUs are suitable for large-scale matrix computing. When the data
scale is small, the performance of the GPU is low, and the communication efficiency between the
CPU and GPU is the biggest bottleneck (Cao et al., 2023). Therefore, the computational
performance of the CAMx-HIP model is not as good as the original Fortran version when MPI
processes increase. According to the characteristics of GPUs suitable for large-scale matrix
computing, the model domain can be expanded, and the model resolution can be increased in the
future to ensure that the amount of data transferred to each GPU reaches the maximum video
memory occupation to make efficient use of GPU. In addition, the advection module only
accounts for about 10% of the total time consumption in the CAMx model (Cao et al., 2023). In
the future, porting the entire integration module except I/O to the GPU is supposed to minimize
the communication frequency.

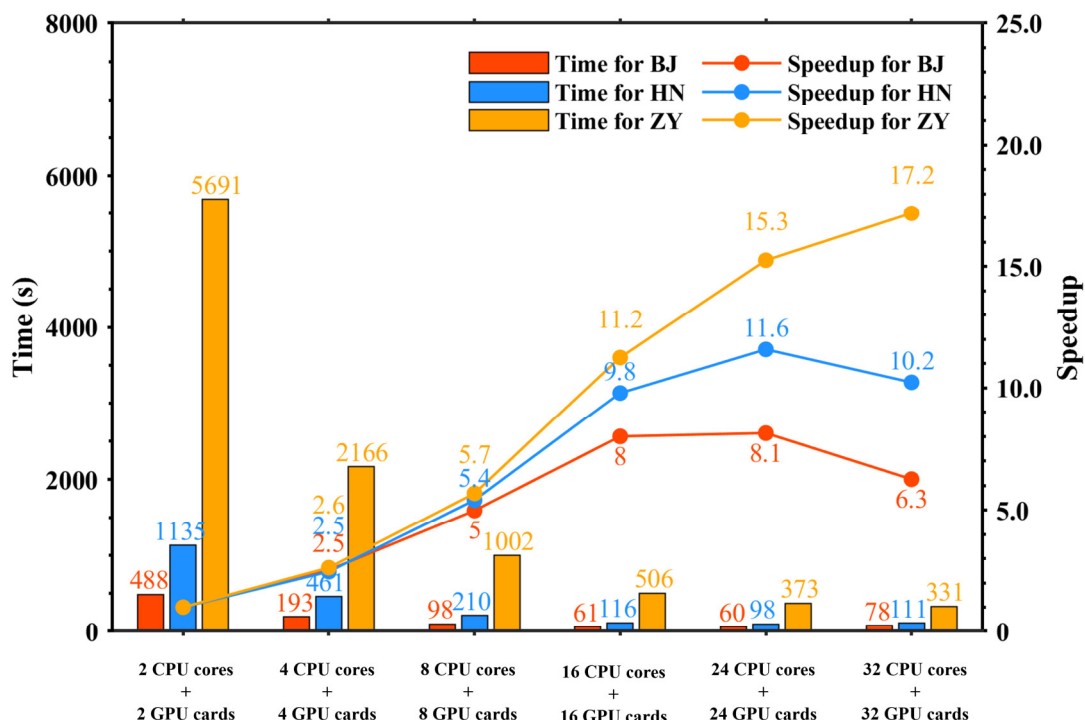


**Figure 9.** The total elapsed time and speedup of the CAMx-HIP model on the "Songshan" supercomputer under the BJ, HN, and ZY cases. The unit is in seconds (s).

The number of GPU accelerators in a single compute node is usually much smaller than the number of CPU cores in heterogeneous HPC systems. Using the hybrid parallel paradigm with MPI and HIP to configure one GPU accelerator for each CPU process results in idle computing resources for the remaining CPU cores. Therefore, the multi-level hybrid parallelism scheme was introduced further to improve the total computational performance of the CAMx-HIP model. As described in the Sect. 3.2, MPI and HIP technology accelerates the horizontal advection module, and the other modules, such as the photolysis module, deposition module, chemical module, etc., which run on the CPU are accelerated by MPI and OMP under the framework of the multi-level hybrid parallelism.

The ZY case achieved the maximum speed-up when launching the 32 domestic CPU processors and GPU-like accelerators. Fig.10 shows the total elapsed time and speedup of CAMx-HIP model in the same configuration when further implementing the multi-level hybrid parallelism on the "Songshan" supercomputer. The AEs of the simulation results between the CAMx-HIP model and CAMx-HIP model with the OMP technology are within ±0.04 ppbV, and the specified results are shown in Figure S1. As the number of threads increases, the elapsed time

of the CAMx-HIP model is further reduced. When a CPU core launches 8 threads, the one-hour

integration time in the CAMx-HIP model has been reduced from 338 seconds to 178 seconds, with

a maximum acceleration of 1.9x.

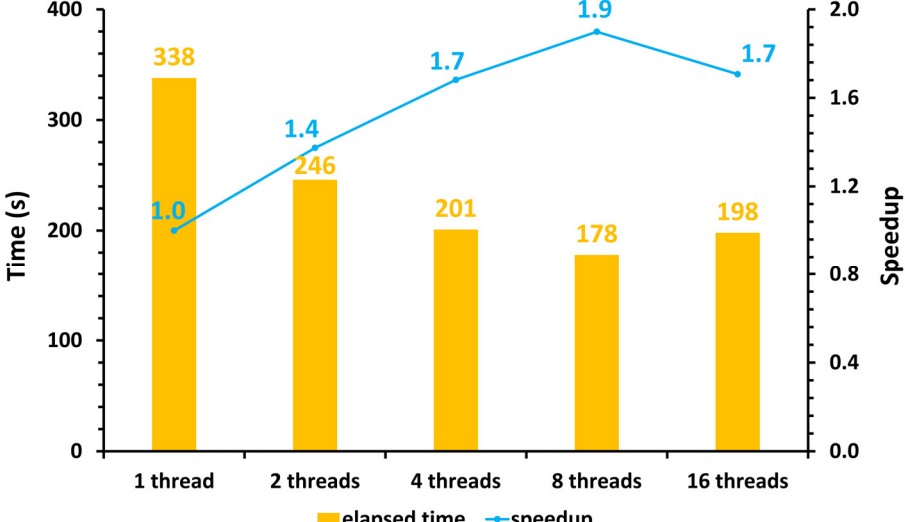

**Figure 10.** The total elapsed time and speedup of the CAMx-HIP model when implementing the multi-level hybrid

parallelism in the ZY case. The unit is in seconds (s).

## 5. Conclusions and discussion

GPUs have become an essential part of providing processing power for high performance

computing applications, especially in geoscience numerical models. Implementing super-large

scale parallel computing of numerical models on GPUs has become one of the significant

directions of its future development. This study implemented the ROCm HIP technology to port

the GPU-HADVPPM from the NVIDIA GPUs to China's domestically GPU-like accelerators.

Further, it introduced the multi-level hybrid parallelism scheme to improve the total computational

performance of the CAMx-HIP model on the China's domestically heterogeneous cluster.

The consistency of model simulation results is a significant prerequisite for heterogeneous

porting. However, the experimental results show that the deviation between the CUDA version

and the Fortran version of the CAMx model, and the deviation between the HIP version and the

Fortran version of the CAMx model, are within the acceptable range, the simulation difference

between the HIP version of CAMx model and Fortran version of CAMx model is more minor. Moreover, the BJ, HN, and ZY test cases can achieve 8.5x, 11.5x, and 28.9x speedup, respectively, when the HADVPPM program is ported from the domestic CPU processor A to the domestic GPU-like accelerator A. The experimental results of different cases show that the larger the computing scale, the obvious more pronounced the acceleration effect of the GPU-HADVPPM program, indicating that GPU is more suitable for super-large scale parallel computing and provides technical support for accurate and fast simulation of ultra-high-resolution air quality at the meter level in the future. The data transfer bandwidth between CPU and GPU is one of the most important factors affecting the computational efficiency of numerical model in heterogeneous clusters, as shown by the fact that the elapsed time of GPU-HADVPPM program on GPU only accounts for 7.3% and 23.8% when considering the data transfer time between CPU and GPU on the the "Songshan" supercomputer and "Taiyuan" computing platform. Therefore, optimizing the data transfer efficiency between CPU and GPU is one of the important directions for the porting and adaptation of geoscience numerical models on heterogeneous clusters in the future.

There is still potential to further improve the computational efficiency of the CAMx-HIP model in the future. First, improve the data transfer efficiency of GPU-HADVPPM between the CPU and the GPU and reduce the data transfer time. Secondly, increase the proportion of HIP C code in CAMx-HIP model on the domestic GPU-like accelerator, and port other modules of CAMx-HIP model to the domestic GPU-like accelerator for computing. Finally, the data type of some variables could be changed from double precision to single precision, and the mixing-precision method is used to further improve the CAMx-HIP computing performance.

*Code and data availability.* The source codes of CAMx version 6.10 are available at https://camx-wp.azurewebsites.net/download/source/ (ENVIRON, 2023). The datasets, the CAMx-HIP codes, as well as the offline test code related to this paper are available online via ZENODO (https://zenodo.org/doi/10.5281/zenodo.10158214), and the CAMx-CUDA code is available online via ZENODO (https://doi.org/10.5281/zenodo.7765218, Cao et al., 2023).

*Author contributions.* KC and QW conducted the simulation and prepared the materials. QW, LiW, and LaW planned and organized the project. KC, QW, HG, HW, XT, and LL refactored and optimized the codes. LiW, NW, HC, DXL, and DQL collected and prepared the data for the simulation. KC, HW, QW, and HG validated and discussed the model results. KC, QW, LiW, NW, XT, HG, and LaW took part in the discussion.

*Competing interests.* The authors declare that they have no conflict of interest.

*Acknowledgements.* The National Key R&D Program of China (grant no. 2020YFA0607804), the National Supercomputing Center in Zhengzhou Innovation Ecosystem Construction Technology Special Program (grant no. 201400210700), GHfund A (grant no. 202302017828), and the Beijing Advanced Innovation Program for Land Surface funded this work. The authors would like to thank the High Performance Scientific Computing Center (HSCC) of Beijing Normal University for providing some high-performance computing environment and technical support.

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
