# Peer review of "GPU-HADVPPM4HIP V1.0: using the heterogeneous interface for portability (HIP) to speed up the piecewise parabolic method in the CAMx (v6.10) air quality model on China's domestic GPU-like accelerator"

_Geoscientific Model Development, 2023_

## Referee Comment (RC2)

[referee-annotated manuscript omitted]

---

## Author Response (AR1)

**Response to Reviewers' comments**

We are thankful to the two reviewers for their thoughtful and constructive comments that help us improve the manuscript substantially. We have revised the manuscript accordingly. Listed below is our point-to-point response in blue to each comment that was offered by the reviewers.

Because experts have raised many questions about the hardware configuration and software environment of the GPU-like accelerator, **Dongxing Li** provided us with a lot of information and gave modification suggestions, so **Dongxing Li** is listed as one of the co-authors in this paper. The author information of this paper is as follows:

*Kai Cao[1], Qizhong Wu[1,5], Lingling Wang[2], Hengliang Guo[3], Nan Wang[2], Huaqiong Cheng[1,5], Xiao Tang[4], Dongxing Li[1,5], Lina Liu[3], Dongqing Li[1], Hao Wu[3], and Lanning Wang[1,5]*

[1]*College of Global Change and Earth System Science, Faculty of Geographical Science, Beijing Normal University, Beijing 100875, China*

[2]*Henan Ecological Environmental Monitoring Centre and Safety Center, Henan Key Laboratory of Environmental Monitoring Technology, Zhengzhou 450008, China*

[3]*National Supercomputing Center in Zhengzhou, Zhengzhou, 450001, China*

[4]*State Key Laboratory of Atmospheric Boundary Layer Physics and Atmospheric Chemistry, Institute of Atmospheric Physics, Chinese Academy of Science, Beijing 100029, China*

[5]*Joint Center for Earth System Modeling and High Performance Computing, Beijing Normal University, Beijing, 100875, China*

**Response to Reviewer #1**

**General Comments:**

The manuscript gmd-2023-222 by Kai Cao et al. describes the porting of the Piecewise Parabolic Method numerical solver as part of the CAMx air quality model on heterogeneous computing architectures, and in particular on the new Chinese domestic accelerators, in comparison to NVIDIA GPUs. The resulting speedup and scaling behaviour are investigated. The study is relevant for the audience of the GMD journal and is timely in the context of the advent of GPU technologies in geoscientific model development.

**Response:** We appreciate the editor for reviewing our manuscript and for the valuable suggestions, which we will address point by point in the following.

The use of English is poor has to be improved throughout the manuscript. The text has to be revised

and edited for syntax and grammar before it is reconsidered for publication.

**Response:** Thanks for the constructive comment. In order to improve the English language of the manuscript, we will call for the professional English language services to polish the manuscript after the public discussion.

**Specific Comments:**

The title is too long and may be confusing to the reader. In particular, it is not clear what "higher" model accuracy is with respect to. Please consider revising to "GPU-HADVPPM4HIP V1.0: using the heterogeneous interface for portability (HIP) to speed up the piecewise parabolic method in the CAMx (v6.10) air quality model on China's domestic GPU-like accelerator"

**Response:** Thanks for the constructive suggestion. We will modify the title according to the suggestion in the revised manuscript.

Lines 53-60: the details of specific supercomputers are not needed and may be supefluous for the reader. Propose to remove these lines.

**Response:** Thanks for the constructive suggestion. We have deleted the details of specific supercomputers in **lines 46-52**, which are as follows:

**Lines 46-52:**

*The 61st edition of the top 10 list, released in June 2023, reveals that 80% of advanced supercomputers adopt the heterogeneous architectures (Top500, 2023), and the Frontier system equipped with AMD Instinct MI250X GPU at the Oak Ridge National Laboratory remains the only true exascale machine with the High-Performance Linpack benchmark (HPL) score of 1.194 Exaflop/s (News, 2023). How to realize the large-scale parallel computing and improve the computational performance of geoscience numerical models on the GPU has become one of the significant directions for the future development of numerical models.*

L143-154: Is there a reference to the architecture of the Chinese heterogeneous cluster? In what ways to generations A and B differ? How do they compare to the NVIDIA clusters? These details are necessary for the reader to understand the potential and achieved performance comparison.

**Response:** Thanks for the constructive suggestion. In this study, two China's domestically heterogeneous clusters newly used in this research, namely "Songshan" supercomputer and "Taiyuan" computing platform. Two domestic heterogeneous clusters both include thousands of computing nodes and each containing one China's domestically CPU processor, four China's domestically GPU-like accelerators, and 128 GB of DDR4 2666 memory. The domestic CPU has

32 X86 based processors. The accelerator adopts a GPU-like architecture consisting of a 16 GB HBM2 device memory and many compute units. The GPU-like accelerators connected to CPU with PCI-E, the peak bandwidth of the data transfer between main memory and device memory is 16 GB/s. The "Taiyuan" computing platform is the next generation of "Songshan" supercomputer, which has been updated in three main aspects: the CPU clock speed has been increased from 2.0 GHz to 2.5 GHz, the number of GPU-like computing units has been increased from 3,840 to 8,192, and the peak bandwidth between main memory and video memory has been increased from 16 GB/s to 32 GB/s. The domestic GPU-like accelerator is similar in architecture to the NVIDIA GPU, and the CUDA core of the NVIDIA GPU essentially corresponds to the computing unit of the domestic GPU. In addition, the NVIDA GPU is programmed using the CUDA toolkit, and the domestic GPU-like is programmed using the ROCm-HIP toolkit (ROCm, 2023). We have modified this part in **lines 127-147**, which are as follows:

**Lines 127-147:**

*Table 1 lists four GPU clusters where we conducted the experiments, two NVIDIA heterogeneous clusters which have the same hardware configuration as Cao et al. (2023) and two China's domestically heterogeneous clusters newly used in this research, namely "Songshan" supercomputer and "Taiyuan" computing platform. Two NVIDIA heterogeneous clusters are equipped with NVIDIA Tesla K40m and V100 GPU accelerators, respectively. Both two domestic clusters include thousands of computing nodes and each containing one China's domestically CPU processor, four China's domestically GPU-like accelerators, and 128 GB of DDR4 2666 memory. The domestic CPU has four NUMA nodes, each NUMA node has eight X86 based processors. The accelerator adopts a GPU-like architecture consisting of a 16 GB HBM2 device memory and many compute units. The GPU-like accelerators connected to CPU with PCI-E, the peak bandwidth of the data transfer between main memory and device memory is 16 GB/s.*

*It is worth noting that the "Taiyuan" computing platform, which has been updated in three main aspects compared to the "Songshan" supercomputer. The CPU clock speed has been increased from 2.0 GHz to 2.5 GHz, the number of GPU-like computing units has been increased from 3,840 to 8,192, and the peak bandwidth between main memory and video memory has been increased from 16 GB/s to 32 GB/s. In terms of the software environment, the NVIDA GPU is programmed using the CUDA toolkit, and the domestic GPU-like is programmed using the ROCm-HIP toolkit developed by AMD (ROCm, 2023). More details about the hardware composition and software environment of the four heterogeneous clusters are presented in Table 1.*

*Table 1. Configurations of the NVIDIA K40m cluster, NVIDIA V100 cluster, "Songshan" supercomputer, and "Taiyuan" computing platform.*

| | Hardware components | |
|---|---|---|
| | *CPU* | *GPU* |
| *NVIDIA K40m cluster* | *Intel Xeon E5-2682 v4 CPU* | *NVIDIA Tesla K40m GPU, 2880 CUDA* |

| | @2.5 GHz, 16 cores | cores, 12 GB video memory |
|---|---|---|
| *NVIDIA V100 cluster* | *Intel Xeon Platinum 8168 CPU @2.7 GHz, 24 cores* | *NVIDIA Tesla V100 GPU, 5120 CUDA cores, 16 GB video memory* |
| *Songshan supercomputer* | *China's domestically CPU processor A, 2.0GHz, 32 cores* | *China's domestically GPU-like accelerator A, 3840 computing units, 16 GB memory* |
| *Taiyuan computing platform* | *China's domestically CPU processor B, 2.5GHz, 32 cores* | *China's domestically GPU-like accelerator B, 8192 computing units, 16 GB memory* |
| | *Software environment* | |
| | *Compiler and MPI* | *Programming model* |
| *NVIDIA K40m cluster* | *Intel Toolkit 2021.4.0* | *CUDA-10.2* |
| *NVIDIA V100 cluster* | *Intel Toolkit 2019.1.144* | *CUDA-10.0* |
| *Songshan supercomputer* | *Intel Toolkit 2021.3.0* | *ROCm-4.0.1/ DTK-23.04* |
| *Taiyuan computing platform* | *Intel Toolkit 2021.3.0* | *DTK-23.04* |

Table 1 is unnecessary and need not be reproduced here. It's enough to refer to the HIP API.

**Response:** Thanks for the constructive suggestion. We have removed the Table 1 and modified this part in **lines 116-124**, which are as follows:

**Lines 116-124:**

*In general, ROCm for the AMD GPU is equivalent to CUDA for NVIDIA GPU. On the ROCm software platform, it uses the AMD's HIP interface which is a C++ runtime API allowing developers to run programs on AMD GPUs. In general, they are very similar and their code can be converted directly by replacing the string "cuda" with "hip" in the most cases. More information about HIP API is available on the AMD ROCm website (ROCm, 2023). Similar to AMD GPU, developers can also use ROCM-HIP programming interface to implement programs running on the China's domestically GPU-like accelerator. The CUDA code cannot run directly on domestic GPU-like accelerators, and it needs to be transcoded into HIP code.*

Figure 1: It is not clear if the trancode happened from CUDA C to HIP C directly ("hipify" in the diagram) or the standard C code was changes (HIP arrow). The section does not document if any changes in the memory size/number of threads/blocks/kernels offloaded were necessary to support the GPU-like accelerator.

**Response:** Sorry for not being able to explain it clearly. There are two ways to implement the conversion of standard C code to HIP C code. Firstly, HIP technology is directly used to convert C code to HIP C code by adding related built-in functions (such as hipMalloc, hipMemcpy, hipFree, etc.). Secondly, based on the existing CUDA C code, the hipify toolkit was used to automatically replace the function names in the CUDA C code with the corresponding names in the HIP C code. The hipify toolkit is essentially a simple script written in the Perl language, and its function is text

replacement, which replaces the function name in CUDA C code with the corresponding name HIP C code according to certain rules. For example, for the memory allocation function cudaMalloc in CUDA, the hipify toolkit can automatically recognize and replace it with hipMalloc. Therefore, the memory size/number of threads/blocks/kernels offloaded do not change when transcoding with the hipify toolkit. We have modified this part in **lines 153-172**, which are as follows:

**Lines 153-172:**

*Fig. 1 shows the heterogeneous porting process of HADVPPM from CPU to NVIDIA GPU and domestic GPU-like accelerator. First, the original Fortran code was refactored using standard C language. Then the CUDA and ROCm-HIP technology were used to convert the standard C code into CUDA C and HIP C code to make it computable on the NIVIDA GPU and domestic GPU-like accelerator. Similar to CUDA technology, the HIP technology is implemented to convert the standard C code to HIP C code by adding related built-in functions (such as hipMalloc, hipMemcpy, hipFree, etc.). To facilitate the portability of applications across different GPU platforms, ROCm provides hipify toolkits to help transcode. The hipify toolkit is essentially a simple script written in the Perl language, and its function is text replacement, which replaces the function name in CUDA C code with the corresponding name in HIP C code according to certain rules. For example, for the memory allocation function cudaMalloc in CUDA, the hipify toolkit can automatically recognize and replace it with hipMalloc. Therefore, the thread and block configuration of GPU remain unchanged due to the simple text substitution during the transcoding. In this study, the ROCm HIP technology was used to implement the operation of GPU-HADVPPM on domestic GPU-like accelerator based on the CUDA version of GPU-HADVPPM which was developed by Cao et al. (2023). The HIP code was compiled using the "hipcc" compiler driver with the library flag "-lamdhip64".*

[Figure]

**Figure 1.** *The heterogeneous porting process of HADVPPM Fortran code from CPU to NVIDIA GPU and domestic GPU-like accelerator.*

In Fig 2, the colour scale for the last two columns is generally too coarse and unsuitable to show any differences, other than a few points. Please consider revising. Are the concentrations for a specific time (for intance at the end of the run when any errors presumably accumulate)? Please give more detail.

**Response:** Thanks for the constructive suggestion. We reduced the color scale in the last two columns of Figure 2, and the absolute errors (AEs) between the Fortran version and CUDA version is more obvious than the HIP and Fortran version. In addition, considering the AEs accumulation and growth, Figure 3 highlights the time series of AEs between Fortran and CUDA versions and between Fortran and HIP versions after grid averaging. As is shown in Figure 3, the AEs of $O_3$, $PSO_4$, CO, and $NO_2$ between the Fortran version and the CUDA version are -0.0002 to 0.0001 ppbV, -0.00003 to 0.00001 $\mu g \cdot m^{-3}$, -0.0004 to 0.0004 ppbV, and -0.0002 to 0.0002 ppbV, respectively, and fluctuate. Although the AEs of the above four species between the Fortran and the HIP version also fluctuates, the fluctuation range is much smaller than that of the CUDA version. Importantly, the AEs between Fortran and CUDA versions and between Fortran and HIP versions both do not accumulate and grow over prolonged simulation periods. We have modified this part in **lines 236-276**, which are as follows:

**Lines 236-276:**

*The hourly concentrations of four major species, i.e. $O_3$, $PSO_4$, CO, and $NO_2$, outputted by the Fortran, CUDA, and HIP versions of CAMx for the BJ case are compared to verify the results correctness before testing the computational performance. Fig. 3 shows the four major species simulation results of the three CAMx version, including Fortran version on the Intel E5-2682 v4 CPU, CUDA version on the NVIDIA K40m cluster and HIP version on the "Songshan" supercomputer, after 48 hours integration, as well as the absolute errors (AEs) of their concentrations. As described by Cao et al. (2023), the parallel design of the CAMx model adopts the primary/secondary mode, and P0 process is responsible for inputting and outputting the data and calling the MPI_Barrier function to synchronize the process, and the other processes are responsible for simulation. When comparing the simulation results, we only launched 2 CPU processes on the CPU platform, and launched 2 CPU processes and configure 2 GPU accelerators on the NVIDIA K40m cluster and "Songshan" supercomputer, respectively.*

*The species' spatial pattern of three CAMx versions on different platform are visually very consistent, and the AEs between the HIP and Fortran version is much smaller than the CUDA and Fortran version. For example, the AEs between the CUDA and Fortran version for $O_3$, $PSO_4$, and $NO_2$ are in the range of $\pm 0.04$ ppbV, $\pm 0.02$ $\mu g \cdot m^{-3}$, and $\pm 0.04$ ppbV. And the AEs between the HIP and Fortran version for above the three species are fall into the range of $\pm 0.01$ ppbV, $\pm 0.005$ $\mu g \cdot m^{-3}$, and $\pm 0.01$ ppbV. For CO, AEs is relatively large due to its high background concentration. However, the AEs between the HIP and Fortran versions is also less than that between the CUDA and Fortran versions where were in the range of $\pm 0.4$ ppbV and $\pm 0.1$ ppbV, respectively.*

*Considering the situation of AEs accumulate and grow, Figure 4 highlights the time series of AEs between Fortran and CUDA versions and between Fortran and HIP versions after grid averaging. As is shown in Fig. 4, the AEs of $O_3$, $PSO_4$, CO, and $NO_2$ between the Fortran version and the CUDA version are -0.0002 to 0.0001 ppbV, -0.00003 to 0.00001 $\mu g \cdot m^{-3}$, -0.0004 to 0.0004 ppbV, and -0.0002 to 0.0002 ppbV, respectively, and fluctuate. Although the AEs of the above four species between the Fortran and the HIP version also fluctuates, the fluctuation range is much smaller than that of the CUDA version. Importantly, the AEs between Fortran and CUDA versions and between Fortran and HIP versions both do not accumulate and grow over prolonged simulation periods.*

[Figure]

***Figure 3.*** *O₃, PSO₄, CO, and NO₂ concentrations outputted by the CAMx Fortran version on the Intel E5-2682 v4 CPU, CUDA version on the NVIDIA K40m cluster and HIP version on the "Songshan" supercomputer under the BJ case. Panels (a), (f), (k), and (p) are from the Fortran version of simulation results for four species. Panels (b), (g), (l), and (q) are from the CUDA version of simulation results for four species. Panels (c), (h), (m), and (r) are from the HIP version of simulation results for four species. Panels (d), (i), (n), and (s) are the AEs between the Fortran and CUDA versions. Panels (e), (j), (o), and (t) are the AEs between the Fortran and HIP versions.*

[Figure]

**Figure 4.** *The time series of AEs between Fortran and CUDA versions (solid blue line) and between Fortran and HIP versions (solid red line) after grid averaging. Panel (a)~(d) represent the AEs of $O_3$, $PSO_4$, CO, and $NO_2$, respectively.*

L. 297-290: What does it mean that the NVIDIA GPU sacrifices part of the accuracy for improved computing performance? Is this for floating point representation and/or arithmetic? Is there no user option (e.g. optimisation levels at compile time) to control the accuracy? Please elaborate.

**Response:** Sorry for not being able to explain it clearly. We have modified the above statement as "NVIDIA GPU loss part of the precision to improve the computing performance". There is a strong correlation between processor performance and precision. In the field of geoscience numerical models, many model optimization works improve the computational performance by reducing the precision of the data, such as Váňa et al. (2017) changed some variables precision in the atmospheric model from double precision to single precision, which increased the overall computational efficiency by 40%, and Wang et al. (2019) improved the computational efficiency of the gas-phase chemistry module in the air quality mode by 25%~28% by modifying the floating-point precision compile flag. In this study, GPU-HADVPPM4HIP has less bias on the domestic GPU-like accelerator. We infer that this may be related to the manufacturing process of NVIDIA GPUs, especially NIVIDA Tesla V100 series GPUs, which may use unknown optimizations to improve GPU performance efficiency by losing part of the accuracy. We also want to know the specific reasons for this, but we are not professional GPU research and development designers after all, so we can only present our experimental results in the air pollution model to you, and discuss with each other to jointly promote the application of GPU in the field of geoscience numerical models. We have modified this part in **lines 317-333**, which are as follows:

**Lines 317-333:**

*From AEs, REs, and ratio of RMSE and std between different CAMx versions, it is less*

*difference that the GPU-HADVPPM4HIP program runs on the "Songshan" supercomputer. Because the simulation accuracy of geoscience numerical model is closely related to the model efficiency, and many model optimization works improve the computational performance by reducing the precision of the data, such as Váňa et al. (2017) changed some variables precision in the atmospheric model from double precision to single precision, which increased the overall computational efficiency by 40%, and Wang et al. (2019) improved the computational efficiency of the gas-phase chemistry module in the air quality mode by 25%~28% by modifying the floating-point precision compile flag. Therefore, we speculate that this may be related to the manufacturing process of NVIDIA GPUs and domestic GPU-like accelerators, especially NIVIDA Tesla V100 series GPUs, which may use unknown optimizations to improve GPU performance efficiency by losing part of the accuracy. In this study, we mainly focus on numerical simulation. Of course, we also want to know the specific reasons for this, but we are not professional GPU research and development designers after all and do not know the underlying design logic of the hardware, so we can only present our experimental results in the air pollution model to you, and discuss with each other to jointly promote the application of GPU in the field of geoscience numerical models.*

Sec. 4.3.1: Given the large time spend to transfer the memory to and from the accelerator, wouldn't a more relevant comparison for any real-world application be of the total required time for the completion of the air quality simulation rather than the compute time of the numerical kernel? Was there any consideration on how to limit the required memory size/bandwidth?

**Response:** Thanks for the constructive suggestion. Table 5 further lists the total elapsed time of CAMx Fortran and HIP versions for BJ case in domestic clusters A and B, and the computing time of advection module with or without data transfer. By coupling the GPU-HADVPPM4HIP to CAMx model and adopting a series of optimizations including communication optimization, memory access optimization, and 2D thread optimization (Cao et al.,2023), the overall computation time of CAMx-HIP model on a single domestic GPU-like accelerator is faster than that of the original Fortran version on a single domestic CPU core. For example, on the "Songshan" supercomputer, one hour of simulation in CAMx-HIP model takes 469 seconds, and the Fortran version takes 481 seconds. In the domestic cluster B, the acceleration effect is more obvious due to the upgrade of hardware and network bandwidth, and the integration time of CAMx-HIP model is 433 seconds when maintaining the same software environment, and the integration time of the Fortran version is 453 seconds. Video memory and bandwidth are the two most significant factors affecting GPU performance, and high video memory and high bandwidth can better play the powerful computing performance of GPUs. Usually, the memory and bandwidth of the GPU are already given at the factory. In this case, the amount of data transferred to the GPU can be roughly estimated before the data is transferred to the GPU, and the amount of data transferred to the GPU can be adjusted according to the size of the GPU memory to ensure that the amount of data transferred to the GPU each time reaches the maximum GPU video memory, so as to give full play to the GPU performance

more efficiently. We have modified this part in **lines 381-439**, which are as follows:

**Lines 381-439:**

[revised manuscript text omitted]

Sec. 4.3.2: Are these results for the total model, or just the accelerated portion? How many timesteps is 1-hour of simulation? It would be generally interesting to see what the average speedup per timestep. It is hard to judge what the overall impact of the accelerator is, given that also the number of CPU cores/processes is also increasing. It would be good to conduct and add a scaling test purely on the CPU cores (ie. without acceleration) to isolate the speedup due to acceleration. Finally, where is the speedup performance saturation (e.g. for the BJ case) attributed for large core/cards counts?

**Response:** Thanks for the constructive suggestion. All the test results in Sec. 4.3.2 are the total elapsed time for one hour simulation. Typically, super-large heterogeneous clusters contain thousands of compute nodes, and one or more GPU accelerators are configured for each compute node. In order to expand the parallel scale of CAMx-HIP model in domestic heterogeneous clusters and improve GPU-like accelerator utilization, we adopt the "MPI+HIP" hybrid parallel scheme, that is, a GPU-like accelerator is configured for each CPU core during the simulation, and the parallel scalability of the three test cases is shown in Figure 5. For the BJ and HN case, the parallel scalability is highest when configured with 24 CPU cores and 24 GPU-like accelerators, with speedup of 8.1x and 11.6x, respectively. In terms of the ZY case, due to its large number of grids, the parallel scalability is the highest when 32 CPU cores and 24 GPU cards are configured, and the acceleration ratio is 17.2x.

Data transfer between CPU and GPU takes several times more time than computation. Regardless of the CPU-GPU data transfer consumption, GPU-HADVPPM4HIP can achieve up to 28.9x speedup on a single domestic GPU-like accelerator. However, in terms of the total time consumption, the CAMx-HIP model is only 10~20 seconds faster than the original Fortran version when one GPU-like accelerator is configured. And as the number of CPU cores and GPU-like accelerators increases, the overall computing performance of CAMx-HIP model is lower than that of the original Fortran version. The main reason is related to the amount of data transferred to GPU. As the number of MPI processes increases, the number of grids responsible for each process decreases, and the amount of data transmitted by the advection module from CPU to GPU decreases. However, GPUs are suitable for large-scale matrix computing. When the data scale is small, the performance of GPU is low, and the communication efficiency between CPU-GPU is the biggest bottleneck (Cao et al., 2023). Therefore, the computational performance of CAMx-HIP model is not as good as the original Fortran version when MPI processes increase. According to the characteristics of GPUs suitable for large-scale matrix computing, the model domain can be expanded and the model resolution can be increased in the future to ensure that the amount of data transferred to each GPU reaches the maximum video memory occupation, so as to make efficient use of GPU. In addition, the advection module only accounts for about 10% of the total time consumption in CAMx model (Cao et al., 2023), and in the future, it is considered to port the entire integration module except I/O to the GPU to minimize the communication frequency.

The timestep of BJ, HN and ZY case were 59, 47, and 61, respectively. Figure 6 shows the GPU-HADVPPM4HIP acceleration in each time step on a single GPU-like accelerator. It can be seen from the figure that all three cases have the smallest speedup of 8.2x, 11.2x, and 27.8x at the first timestep, which is related to the time required for GPU-like accelerator startup. When the GPU-like is started and operating normally, the speedup of the three cases tend to be stable in the following time steps, and stabilize around 8.5x, 11.5x and 28.0x respectively. We have modified this part in **lines 372-380** and **lines 441-474**, which are as follows:

**Lines 372-380:**

The timestep of BJ, HN and ZY case were 59, 47, and 61, respectively. Fig. 7 shows the GPU-HADVPPM4HIP acceleration in each time step on a single domestic GPU-like accelerator A. It can be seen from the figure that all three cases have the smallest speedup of 8.2x, 11.2x, and 27.8x at the first timestep, which is related to the time required for GPU-like accelerator startup. When the GPU-like is started and operating normally, the speedup of the three cases tend to be stable in the following time steps, and stabilize around 8.5x, 11.5x and 28.0x respectively.

[Figure]

**Figure 7.** The GPU-HADVPPM4HIP acceleration in each time step on a single GPU-like accelerator for BJ, HN, and ZY case. The timestep of above three cases are 59, 47, and 61, respectively.

**Lines 441-474:**

Generally, the heterogeneous HPC systems have thousands of compute nodes which are equipped with one or more GPUs on each compute node. To make full use of multiple GPUs, a parallel architecture with an MPI and CUDA hybrid paradigm was implemented to improve the overall computational performance of CAMx-CUDA model (Cao et al., 2023). In this studying, the hybrid parallelism with an MPI and HIP paradigm was used to implement the HIP version of GPU-HADVPPM run on multiple domestic GPU-like accelerators. Fig.8 shows the total elapsed time and speedup of CAMx-HIP model which coupled with the HIP version GPU-HADVPPM on the "Songshan" supercomputer under the BJ, HN, and ZY cases. The simulation of above three cases for one hour took 488 seconds, 1135 seconds and 5691 seconds respectively when launching two domestic CPU processors and two GPU-like accelerators. For the BJ and HN case, the parallel scalability is highest when configured with 24 CPU cores and 24 GPU-like accelerators, with speedup of 8.1x and 11.6x, respectively. In terms of the ZY case, due to its large number of grids, the parallel scalability is the highest when 32 CPU cores and 32 GPU-like accelerators are configured, and the acceleration ratio is 17.2x.

As mentioned above, data transfer between CPU and GPU takes several times more time than computation. Regardless of the CPU-GPU data transfer consumption, GPU-HADVPPM4HIP can achieve up to 28.9x speedup on a single domestic GPU-like accelerator. However, in terms of the

*total time consumption, the CAMx-HIP model is only 10~20 seconds faster than the original Fortran version when one GPU-like accelerator is configured. And as the number of CPU cores and GPU-like accelerators increases, the overall computing performance of CAMx-HIP model is lower than that of the original Fortran version. The main reason is related to the amount of data transferred to GPU. As the number of MPI processes increases, the number of grids responsible for each process decreases, and the amount of data transmitted by the advection module from CPU to GPU decreases. However, GPUs are suitable for large-scale matrix computing. When the data scale is small, the performance of GPU is low, and the communication efficiency between CPU and GPU is the biggest bottleneck (Cao et al., 2023). Therefore, the computational performance of CAMx-HIP model is not as good as the original Fortran version when MPI processes increase. According to the characteristics of GPUs suitable for large-scale matrix computing, the model domain can be expanded and the model resolution can be increased in the future to ensure that the amount of data transferred to each GPU reaches the maximum video memory occupation, so as to make efficient use of GPU. In addition, the advection module only accounts for about 10% of the total time consumption in CAMx model (Cao et al., 2023), and in the future, it is considered to port the entire integration module except I/O to the GPU to minimize the communication frequency.*

**Minor Comments:**

L70: Kinesthetic PreProcessor: Accelerated (KPPA) -> It should read "Kinetic PreProcessor"

**Response:** Sorry for this mistake. We have modified the relevant statement in **lines 62-64**, which are as follows:

**Lines 62-64:** *Linford et al. (2011) presented the Kinetic PreProcessor (KPP) to generate the chemical mechanism code in CUDA language which can be implemented on NVIDIA Tesla C1060 GPU.*

L. 115: CUDA is not only supported on Tesla, but all NVIDIA GPU architectures. Please rephrase

**Response:** Sorry for this mistake. We have modified the relevant statement in **lines 109-110**, which are as follows:

**Lines 109-110:** *CUDA is a proprietary application programming interface (API) and as such is only supported on NVIDIA's GPUs.*

L155-160: Most of the technical information is repeated in Table 2 and can be omitted from the text.

**Response:** Thanks for the constructive suggestion. We have omitted the repeated relevant description of Table 2 in **lines 127-147**, which are as follows:

*Table 1 lists four GPU clusters where we conducted the experiments, two NVIDIA heterogeneous clusters which have the same hardware configuration as Cao et al. (2023) and two China's domestically heterogeneous clusters newly used in this research, namely "Songshan" supercomputer and "Taiyuan" computing platform. Two NVIDIA heterogeneous clusters are equipped with NVIDIA Tesla K40m and V100 GPU accelerators, respectively. Both two domestic clusters include thousands of computing nodes and each containing one China's domestically CPU processor, four China's domestically GPU-like accelerators, and 128 GB of DDR4 2666 memory. The domestic CPU has four NUMA nodes, each NUMA node has eight X86 based processors. The accelerator adopts a GPU-like architecture consisting of a 16 GB HBM2 device memory and many compute units. The GPU-like accelerators connected to CPU with PCI-E, the peak bandwidth of the data transfer between main memory and device memory is 16 GB/s.*

*It is worth noting that the "Taiyuan" computing platform, which has been updated in three main aspects compared to the "Songshan" supercomputer. The CPU clock speed has been increased from 2.0 GHz to 2.5 GHz, the number of GPU-like computing units has been increased from 3,840 to 8,192, and the peak bandwidth between main memory and video memory has been increased from 16 GB/s to 32 GB/s. In terms of the software environment, the NVIDA GPU is programmed using the CUDA toolkit, and the domestic GPU-like is programmed using the ROCm-HIP toolkit developed by AMD (ROCm, 2023). More details about the hardware composition and software environment of the four heterogeneous clusters are presented in Table 1.*

***Table 1.*** *Configurations of the NVIDIA K40m cluster, NVIDIA V100 cluster, "Songshan" supercomputer, and "Taiyuan" computing platform.*

| | Hardware components | |
|---|---|---|
| | *CPU* | *GPU* |
| *NVIDIA K40m cluster* | *Intel Xeon E5-2682 v4 CPU @2.5 GHz, 16 cores* | *NVIDIA Tesla K40m GPU, 2880 CUDA cores, 12 GB video memory* |
| *NVIDIA V100 cluster* | *Intel Xeon Platinum 8168 CPU @2.7 GHz, 24 cores* | *NVIDIA Tesla V100 GPU, 5120 CUDA cores, 16 GB video memory* |
| *Songshan supercomputer* | *China's domestically CPU processor A, 2.0GHz, 32 cores* | *China's domestically GPU-like accelerator A, 3840 computing units, 16 GB memory* |
| *Taiyuan computing platform* | *China's domestically CPU processor B, 2.5GHz, 32 cores* | *China's domestically GPU-like accelerator B, 8192 computing units, 16 GB memory* |
| | Software environment | |
| | *Compiler and MPI* | *Programming model* |
| *NVIDIA K40m cluster* | *Intel Toolkit 2021.4.0* | *CUDA-10.2* |
| *NVIDIA V100 cluster* | *Intel Toolkit 2019.1.144* | *CUDA-10.0* |
| *Songshan supercomputer* | *Intel Toolkit 2021.3.0* | *ROCm-4.0.1/ DTK-23.04* |
| *Taiyuan computing platform* | *Intel Toolkit 2021.3.0* | *DTK-23.04* |

Please move all links to websites to the references section

**Response:** Sorry for this mistake. We have moved all links to the references section, which are as follows:

**Lines 46-50:** *The 61st edition of the top 10 list, released in June 2023, reveals that 80% of advanced supercomputers adopt the heterogeneous architectures (Top500, 2023), and the Frontier system equipped with AMD Instinct MI250X GPU at the Oak Ridge National Laboratory remains the only true exascale machine with the High-Performance Linpack benchmark (HPL) score of 1.194 Exaflop/s (News, 2023).*

**Lines 87-90:** *The Comprehensive Air Quality Model with Extensions version 6.10 (CAMx v6.10; ENVIRON, 2014) is a state-of-the-art air quality model which simulates the emission, dispersion, chemical reaction, and removal of the air pollutants on a system of nested three-dimensional grid boxes (CAMx, 2023).*

**Lines 120-121:** *More information about HIP API is available on the AMD ROCm website (ROCm, 2023).*

**Response to Reviewer #2**

**General Comments:**

The paper deals with a comparison of non-accelerated and accelerated air quality simulations, using Intel CPUs, Nvidia GPUs, and unspecified Chinese hardware. The topic is well suited and is overall interesting and relevant. Nonetheless, there are in my opinion major issues with the manuscript which limit its broader interest. I summarise my overall impression and critique here, and also attach an annotated pdf.

**Response:** We appreciate the referee for reviewing our manuscript and for the valuable suggestions, which we will address point by point in the following and improve our revised manuscript according the comments and suggestions in the pdf.

**Q1:** A first major issue in the paper is that the Chinese hardware remains unknown for the readers throughout the text. Very little information about this hardware is disclosed, which makes it very difficult to assess and understand potentially relevant differences. The reader might infer/guess some parts of this (e.g., that CUDA cannot be used to program the Chinese accelerators), but these properties of the hardware are not mentioned. The reason for the lack of information is also not disclosed. In comparison of course, one can find full technical specifications for the Intel and Nvidia hardware used in the study. We are not explained why these Chinese accelerator are GPU-like and not quite GPUs. We are also provided very little information on the Chinese CPUs. This creates, in

my opinion quite some intransparency.

**Response:** Thanks for the constructive comment. Sorry we didn't give more information about the domestic GPUs and domestic CPUs in the paper, and in the revised manuscript we will try our best to improve those information as we can. It maybe that due to the U. S. ban on the sale of some advanced chips to China, and the manufacturers are not willing to release the full technical specifications, so we cannot directly obtain the full information about the GPUs and CPUs. In order to provide the readers with more detailed information, we tried our best to query the software environment and hardware configuration of the clusters by remote access, and listed them in **lines 126-146** of the manuscript, hoping to be helpful to readers. In addition, we reply to the issue that CUDA cannot be used to program on domestic GPUs as follows. At present, the Radeon Open Compute platform (ROCm) toolkit developed by AMD can be used to program the domestic GPU. ROCm is an open-source software platform for HPC and hyperscale GPU computing (ROCm, 2023). On the ROCm software platform, it uses the AMD's HIP interface which is a C++ runtime API to allows developers to run programs on AMD GPUs and Chinese GPU-like accelerators.

**Lines 126-146:**

*Table 1 lists four GPU clusters where we conducted the experiments, two NVIDIA heterogeneous clusters which have the same hardware configuration as Cao et al. (2023) and two China's domestically heterogeneous clusters newly used in this research, namely "Songshan" supercomputer and "Taiyuan" computing platform. Two NVIDIA heterogeneous clusters are equipped with NVIDIA Tesla K40m and V100 GPU accelerators, respectively. Both two domestic clusters include thousands of computing nodes and each containing one China's domestically CPU processor, four China's domestically GPU-like accelerators, and 128 GB of DDR4 2666 memory. The domestic CPU has four NUMA nodes, each NUMA node has eight X86 based processors. The accelerator adopts a GPU-like architecture consisting of a 16 GB HBM2 device memory and many compute units. The GPU-like accelerators connected to CPU with PCI-E, the peak bandwidth of the data transfer between main memory and device memory is 16 GB/s.*

*It is worth noting that the "Taiyuan" computing platform, which has been updated in three main aspects compared to the "Songshan" supercomputer. The CPU clock speed has been increased from 2.0 GHz to 2.5 GHz, the number of GPU-like computing units has been increased from 3,840 to 8,192, and the peak bandwidth between main memory and video memory has been increased from 16 GB/s to 32 GB/s. In terms of the software environment, the NVIDA GPU is programmed using the CUDA toolkit, and the domestic GPU-like is programmed using the ROCm-HIP toolkit developed by AMD (ROCm, 2023). More details about the hardware composition and software environment of the four heterogeneous clusters are presented in Table 1.*

*__Table 1.__ Configurations of the NVIDIA K40m cluster, NVIDIA V100 cluster, "Songshan" supercomputer, and "Taiyuan" computing platform.*

| Hardware components |
| --- |

|  | CPU | GPU |
|---|---|---|
| *NVIDIA K40m cluster* | *Intel Xeon E5-2682 v4 CPU @2.5 GHz, 16 cores* | *NVIDIA Tesla K40m GPU, 2880 CUDA cores, 12 GB video memory* |
| *NVIDIA V100 cluster* | *Intel Xeon Platinum 8168 CPU @2.7 GHz, 24 cores* | *NVIDIA Tesla V100 GPU, 5120 CUDA cores, 16 GB video memory* |
| *Songshan supercomputer* | *China' s domestically CPU processor A, 2.0GHz, 32 cores* | *China's domestically GPU-like accelerator A, 3840 computing units, 16 GB memory* |
| *Taiyuan computing platform* | *China' s domestically CPU processor B, 2.5GHz, 32 cores* | *China's domestically GPU-like accelerator B, 8192 computing units, 16 GB memory* |

| | *Software environment* | |
|---|---|---|
| | *Compiler and MPI* | *Programming model* |
| *NVIDIA K40m cluster* | *Intel Toolkit 2021.4.0* | *CUDA-10.2* |
| *NVIDIA V100 cluster* | *Intel Toolkit 2019.1.144* | *CUDA-10.0* |
| *Songshan supercomputer* | *Intel Toolkit 2021.3.0* | *ROCm-4.0.1/ DTK-23.04* |
| *Taiyuan computing platform* | *Intel Toolkit 2021.3.0* | *DTK-23.04* |

**Q2:** Moreover, the authors make bold claims in terms of the comparative accuracy of Nvidia GPUs, relative to that obtained with the Chinese accelerators (larger errors for Nvidia hardware, which the authors attribute to the fact that Nvidia favours performance over accuracy!). To sustain such claims not only is more evidence required, but also significant information on the hardware and the software stack over which the application is built.

**Response:** Thanks for the constructive comment. In this manuscript, by comparing the simulation results of different versions of CAMx model, it can be seen that GPU-HADVPPM4HIP has less deviation on domestic GPU-like accelerators. Because the simulation accuracy of geoscience numerical model is closely related to the model efficiency, and many model optimization works improve the computational performance by reducing the precision of the data, such as Váňa et al. (2017) changed some variables precision in the atmospheric model from double precision to single precision, which increased the overall computational efficiency by 40%, and Wang et al. (2019) improved the computational efficiency of the gas-phase chemistry module in the air quality mode by 25%~28% by modifying the floating-point precision compile flag. Therefore, we speculate that this may be related to the manufacturing process of NVIDIA GPUs and domestic GPU-like accelerators, especially NIVIDA Tesla V100 series GPUs, which may use unknown optimizations to improve GPU performance efficiency by losing part of the accuracy. In this study, we mainly focus on numerical simulation. Of course, we also want to know the specific reasons for this, but we are not professional GPU research and development designers, and do not know the underlying design logic of the hardware, so we can only present our experimental results in the air pollution model to the readers, and discuss with each other to jointly promote the application of GPU in the field of geoscience numerical models. We have added this discussion in **lines 317-333**, which are as follows:

*From AEs, REs, and ratio of RMSE and std between different CAMx versions, it is less difference that the GPU-HADVPPM4HIP program runs on the "Songshan" supercomputer. Because the simulation accuracy of geoscience numerical model is closely related to the model efficiency, and many model optimization works improve the computational performance by reducing the precision of the data, such as Váňa et al. (2017) changed some variables precision in the atmospheric model from double precision to single precision, which increased the overall computational efficiency by 40%, and Wang et al. (2019) improved the computational efficiency of the gas-phase chemistry module in the air quality mode by 25%~28% by modifying the floating-point precision compile flag. Therefore, we speculate that this may be related to the manufacturing process of NVIDIA GPUs and domestic GPU-like accelerators, especially NIVIDA Tesla V100 series GPUs, which may use unknown optimizations to improve GPU performance efficiency by losing part of the accuracy. In this study, we mainly focus on numerical simulation. Of course, we also want to know the specific reasons for this, but we are not professional GPU research and development designers after all and do not know the underlying design logic of the hardware, so we can only present our experimental results in the air pollution model to you, and discuss with each other to jointly promote the application of GPU in the field of geoscience numerical models.*

**Q3:** My second point is that the manuscript is a bit confusing in terms of which processes are mapped to which hardware. This may be simply a matter of clarification, in particular when it comes to the in-node heterogeneous processes ("other modules" the authors state, are solved using OpenMP on the CPUs, whereas the advection module, when solved on CPUs is on a single core?). I would suggest a sketch to explain this.

**Response:** Sorry for not being able to explain it clearly and thanks for the constructive comment. The original CAMx model supports message passing interface (MPI) parallel technology running on the general-purpose CPU. The simulation domain is divided into several sub-regions by MPI, and each CPU process is responsible for simulation of its sub-region, which includes advection module and other modules such as photolysis module, deposition module, chemical module, etc. In the previous study, Cao et al. (2023) adopt a parallel architecture with an MPI and CUDA (MPI+CUDA) hybrid paradigm to configure one GPU accelerator for each CPU process. For the advection module, the simulation originally implemented by the CPU is handed over to the GPU. Other module computing tasks continue to be completed on the CPU. In this study, in addition to the MPI+HIP hybrid parallel scheme to realize the advection module running on several domestic GPU-like accelerators, the OpenMP (OMP) hybrid parallel framework of CAMx model is further introduced to realize the MPI+OMP hybrid parallelism of other modules on CPU. A schematic of the multi-level hybrid parallel framework is shown in Figure 2. For example, in a computing node, four CPU processes and four GPU-like accelerators are launched, and each CPU process spawns

four threads. Then the advection module is simulated by 4 GPU-like accelerators, and the other modules are done by 4*4 threads spawned by CPU processes. We have modified this part in **lines 183-204**, which are as follows:

**Lines 183-204:**

*As mentioned above, the original CAMx model supports message passing interface (MPI) parallel technology running on the general-purpose CPU. The simulation domain is divided into several sub-regions by MPI, and each CPU process is responsible for computation of its sub-region, which includes the computation tasks of advection module and other modules such as photolysis module, deposition module, chemical module, etc. In the previous studying, Cao et al. (2023) adopt a parallel architecture with an MPI and CUDA (MPI+CUDA) hybrid paradigm to configure one GPU accelerator for each CPU process. For the advection module, the simulation originally implemented by the CPU is handed over to the GPU. Other module computing tasks continue to be completed on the CPU.*

*In this study, when the CUDA C code of GPU-HADVPPM is converted to HIP C code, GPU-HADVPPM with an MPI and HIP (MPI+HIP) heterogeneous hybrid programming technology can also run on multiple domestic GPU-like accelerators. However, the number of GPU-like accelerators in a single compute node is usually much smaller than the number of CPU cores in the heterogeneous HPC systems. Therefore, in order to make full use of the remaining CPU computing resources, the OMP API of CAMx model is further introduced to realize the MPI+OMP hybrid parallelism of other modules on CPU. A schematic of the multi-level hybrid parallel framework is shown in Figure 2. For example, in a computing node, four CPU processes and four GPU-like accelerators are launched, and each CPU process spawns four threads. Then the advection module is simulated by 4 GPU-like accelerators, and the other modules are done by 4*4 threads spawned by CPU processes.*

[Figure]

**Figure 2.** *A schematic of the multi-level hybrid parallel framework.*

**Q4:** In terms of the results, the text needs to be much more specific on how the runs are computed and how the speed up is computed. One can infer that speed up is computed against runs on the Chinese CPU, but it is not fully clear if that is using a parallel run on all cores or on a serial job.

**Response:** Sorry for not being able to explain it clearly. According to the results of Cao et al. (2023), the parallel design of the CAMx model adopts the primary/secondary mode, and Process 0 (P0) is responsible for inputting and outputting the data and calling the MPI_Barrier function to synchronize the process, and the other processes are responsible for simulation. When testing the computational performance of the advection module on the CPU, we only launch two CPU processes, namely P0 and Process 1 (P1), where P0 is mainly responsible for data input and output and synchronization process, and P1 is mainly responsible for simulation. In the P1 process, the system_clock functions in the Fortran language are used to test the elapsed time of the advection module on the CPU. Similarly, when testing the computation performance of the advection module on the GPU-like accelerator, we only launch 2 CPU processes and 2 GPU-like accelerators. When a P1 process runs to the advection module, the original computation process is migrated from the CPU to the GPU, and the hipEvent_t function in HIP programming is used to test the running time of the advection module on the GPU-like accelerator. We have modified this part in **lines 242-247**, which are as follows:

**Lines 242-247:**

*As described by Cao et al. (2023), the parallel design of the CAMx model adopts the primary/secondary mode, and P0 process is responsible for inputting and outputting the data and calling the MPI_Barrier function to synchronize the process, and the other processes are responsible for simulation. When comparing the simulation results, we only launched 2 CPU processes on the CPU platform, and launched 2 CPU processes and configure 2 GPU accelerators on the NVIDIA K40m cluster and "Songshan" supercomputer, respectively.*

**Q5:** The authors identify host-device transfers as the key bottleneck, mainly by computing the share of time spent on kernels and on data transfers. The authors seem to conclude that this can only be improved with better bandwidth between host and device. However, nothing is said about potential implementation issues (e.g., poor handling of memory allocations) or even bottlenecks which could be alleviated with better suited algorithms. Indeed, the fraction of time the solver actually spends on running GPU kernels is very low (below 24%) which is very inefficient.

**Response:** Sorry for not being able to explain it clearly. Data transfer between CPU and GPU is one of the main factors that limit the efficiency of geoscience numerical models in heterogeneous clusters (Mielikainen et al., 2012; Mielikainen et al., 2013; Huang et al., 2013). On the one hand, improving the data transfer bandwidth between CPU and GPU can improve the computational efficiency of the model in heterogeneous clusters. On the other hand, the optimization measures can be implemented to improve the data transfer efficiency between CPU and GPU. For example, (1) Asynchronous data transfer is used to reduce the communication latency between CPU and GPU. Computation and data transfer are performed simultaneously to hide communication overhead; (2) Currently, some advanced GPU architectures support a unified memory architecture, so that the CPU and GPU can share the same memory space and avoid frequent data transfers. This reduces the overhead of data transfer and improves data transfer efficiency; (3) Cao et al. (2023) adopted communication optimization measures to reduce the communication frequency in one time integration step to one, but there is still the problem of high communication frequency in the whole simulation. In the future, we will consider porting other hotspots of CAMx model, or even the whole integral module except I/O, to GPU-like accelerators for increasing the proportion of code on the GPU and reduce the frequency of CPU-GPU communication. We have added this part in **lines 412-428**, which are as follows:

**Lines 412-428:**

*By comparing the kernel execution time and total running time of GPU-HADVPPM4HIP on the domestic accelerator, it can be seen that the data transfer efficiency between CPU and GPU is really inefficient, which seriously restricts the computational performance of numerical models in heterogeneous clusters. On the one hand, improving the data transfer bandwidth between CPU and GPU can improve the computational efficiency of the model in heterogeneous clusters. On the other hand, the optimization measures can be implemented to improve the data transfer efficiency between*

*CPU and GPU. For example, (1) Asynchronous data transfer is used to reduce the communication latency between CPU and GPU. Computation and data transfer are performed simultaneously to hide communication overhead; (2) Currently, some advanced GPU architectures support a unified memory architecture, so that the CPU and GPU can share the same memory space and avoid frequent data transfers. This reduces the overhead of data transfer and improves data transfer efficiency; (3) Cao et al. (2023) adopted communication optimization measures to reduce the communication frequency in one time integration step to one, but there is still the problem of high communication frequency in the whole simulation. In the future, we will consider porting other hotspots of CAMx model, or even the whole integral module except I/O, to GPU-like accelerators for increasing the proportion of code on the GPU and reduce the frequency of CPU-GPU communication.*

**Q6:** Going back to the accuracy aspect, the authors don't really provide an in-depth discussion of why there are differences between the different computations. The magnitude of the errors is way above arithmetic accuracy of the hardware involved, and the claims that Nvidia hardware favours performance over accuracy are not well supported by the exercise carried out in this manuscript. This requires much better defined benchmarks. Overall, the differences between the simulations could be due to a wide variety of reasons, anywhere from arithmetic accuracy of the hardware and the hardware-specific arithmetic kernels, sure, but all the way up to errors in the different implementations. The investigation is simply not deep enough to support the claims nor to provide robust evidence of the root causes of the differences. There is no discussion on why, for example, the different species show different variability in the errors, although this suggests that whatever the root cause of the error, this propagates differently across different processes/state variables.

**Response:** Thanks for the constructive comment. In the error analysis of this paper, the simulation results of CAMx model on a general CPU processor (Intel Xeon E5-2682 v4 CPU) are used as the benchmark to compare the deviation of simulation results on different GPUs. From the comparison results, GPU-HADVPPM4HIP has less deviation on domestic GPU-like accelerators. As mentioned above, because the simulation accuracy of geoscience numerical model is closely related to the model efficiency, and many model optimization works improve the computational performance by reducing the precision of the data. Therefore, we speculate that this may be related to the manufacturing process of NVIDIA GPUs and domestic GPU-like accelerators, especially NIVIDA Tesla V100 series GPUs, which may use unknown optimizations to improve GPU performance efficiency by losing part of the accuracy. Since we are not GPU developers and do not know the underlying design logic of the hardware, so we can only present our experimental results for discussion. We have added this discussion in **lines 317-333**, which are as follows:

**Lines 317-333:**

*From AEs, REs, and ratio of RMSE and std between different CAMx versions, it is less*

*difference that the GPU-HADVPPM4HIP program runs on the "Songshan" supercomputer. Because the simulation accuracy of geoscience numerical model is closely related to the model efficiency, and many model optimization works improve the computational performance by reducing the precision of the data, such as Váňa et al. (2017) changed some variables precision in the atmospheric model from double precision to single precision, which increased the overall computational efficiency by 40%, and Wang et al. (2019) improved the computational efficiency of the gas-phase chemistry module in the air quality mode by 25%~28% by modifying the floating-point precision compile flag. Therefore, we speculate that this may be related to the manufacturing process of NVIDIA GPUs and domestic GPU-like accelerators, especially NIVIDA Tesla V100 series GPUs, which may use unknown optimizations to improve GPU performance efficiency by losing part of the accuracy. In this study, we mainly focus on numerical simulation. Of course, we also want to know the specific reasons for this, but we are not professional GPU research and development designers after all and do not know the underlying design logic of the hardware, so we can only present our experimental results in the air pollution model to you, and discuss with each other to jointly promote the application of GPU in the field of geoscience numerical models.*

**Below are responses to the comments and suggestions listed in the pdf.**

**Line 2:**

domestic is always relative.

It is clear this refers to the Chinese hardware discussed in the paper. However, the term is found too often in the text and is not particularly useful.

Is there no name for the accelerator? In the text the authors can extend in explaining what this hardware is, including that it is developed in China.

**Response:** Thanks for the constructive comment. On the one hand, Chinese chip manufacturers are reluctant to release specific details of Chinese hardware, possibly due to the impact of the U. S. ban on the sale of advanced GPUs in China. On the other hand, according to Reviewer 1's suggestion, we have changed the title of this paper to: *GPU-HADVPPM4HIP V1.0: using the heterogeneous interface for portability (HIP) to speed up the piecewise parabolic method in the CAMx (v6.10) air quality model on China's domestic GPU-like accelerator.*

**Line 63:**

Strike out

**Response:** Sorry for this mistake. We have revised this part in **lines 53-56**, which are as follows:

**Lines 53-56**: *In terms of the heterogeneous porting for air quality model, most scholars select the chemical module, one of the hotspots, to implement heterogeneous porting, and porting the computational process originally on the CPU processes to the GPU accelerator, in order to improve the computing efficiency.*

**Line 64:**

it is not quite clear what is meant here

**Response:** Sorry for not being able to explain it clearly. In terms of the heterogeneous porting for air quality model, most scholars select the chemical module, one of the hotspots, to implement heterogeneous porting, and porting the computational process originally on the CPU processes to the GPU accelerator, in order to improve the computing efficiency. We have modified this part in **lines 53-56**, which are as follows:

**Lines 53-56**: *In terms of the heterogeneous porting for air quality model, most scholars select the chemical module, one of the hotspots, to implement heterogeneous porting, and porting the computational process originally on the CPU processes to the GPU accelerator, in order to improve the computing efficiency.*

**Line 66:**

compared to what?

**Response:** Sorry for not being able to explain it clearly. Sun et al. (2018) used CUDA technology to port the second-order Rosenbrock solver of chemistry module of CAM4-Chem to NVIDIA Tesla K20X GPU, and achieved up 11.7x speedup compared to the AMD Opteron™ 6274 (Interlagos) CPU (16 cores) using one CPU core. We have modified this part in **lines 56-59**, which are as follows:

**Lines 56-59**: *Sun et al. (2018) used CUDA technology to port the second-order Rosenbrock solver of chemistry module of CAM4-Chem to NVIDIA Tesla K20X GPU, and achieved up 11.7x speedup compared to the AMD Opteron™ 6274 (Interlagos) CPU (16 cores) using one CPU core.*

**Line 73:**

compared to what?

**Response:** Sorry for not being able to explain it clearly. The KPP-generated SAPRC'99 mechanism from CMAQ model achieved a maximum speedup of 13.7x and KPP-generated RADM2 mechanism from WRF-chem model achieved an 8.5x speedup both compared to the Intel Quad-Core Xeon 5400 series CPU. We have modified this part in **lines 64-67**, which are as follows:

**Lines 64-67**: *The KPP-generated SAPRC'99 mechanism from CMAQ model achieved a maximum*

*speedup of 13.7x and KPP-generated RADM2 mechanism from WRF-chem model achieved an 8.5x speedup both compared to the Intel Quad-Core Xeon 5400 series CPU.*

**Line 76:**

Reduction

**Response:** Sorry for this mistake. We have revised this part in **lines 67-72**, which are as follows:

**Lines 67-72:** *Similarly, the advection module is also one of the hotspot modules in the air quality model, Cao et al. (2023) adopted the Fortran-C-CUDA C scheme and implemented a series of optimizations, including reduction the CPU–GPU communication frequency, optimize the GPU memory access, and thread and block co-indexing, to increase the computational efficiency of the HADVPPM advection solver. It can achieve up to the 18.8x speedup on the NVIDIA Tesla V100 GPU compared to the Intel Xeon Platinum 8168 CPU.*

**Line 75:**

this clause here is hanging. I guess you intend to say something like "Conernign horizontal advection modules for atmospheric chemical models, Cao..."

**Response:** Sorry for not being able to explain it clearly. Similarly, the advection module is also one of the hotspot modules in the air quality model, Cao et al. (2023) adopted the Fortran-C-CUDA C scheme and implemented a series of optimizations, including reduction the CPU–GPU communication frequency, optimize the GPU memory access, and thread and block co-indexing, to increase the computational efficiency of the HADVPPM advection solver. It can achieve up to the 18.8x speedup on the NVIDIA Tesla V100 GPU compared to the Intel Xeon Platinum 8168 CPU. We have revised this part in **lines 67-72**, which are as follows:

**Lines 67-72:** *Similarly, the advection module is also one of the hotspot modules in the air quality model, Cao et al. (2023) adopted the Fortran-C-CUDA C scheme and implemented a series of optimizations, including reduction the CPU–GPU communication frequency, optimize the GPU memory access, and thread and block co-indexing, to increase the computational efficiency of the HADVPPM advection solver. It can achieve up to the 18.8x speedup on the NVIDIA Tesla V100 GPU compared to the Intel Xeon Platinum 8168 CPU.*

**Line 78:**

compared to what?

**Response:** Sorry for not being able to explain it clearly. Similarly, the advection module is also one of the hotspot modules in the air quality model, Cao et al. (2023) adopted the Fortran-C-CUDA C

scheme and implemented a series of optimizations, including reduction the CPU–GPU communication frequency, optimize the GPU memory access, and thread and block co-indexing, to increase the computational efficiency of the HADVPPM advection solver. It can achieve up to the 18.8x speedup on the NVIDIA Tesla V100 GPU compared to the Intel Xeon Platinum 8168 CPU. We have revised this part in **lines 67-72**, which are as follows:

**Lines 67-72:** *Similarly, the advection module is also one of the hotspot modules in the air quality model, Cao et al. (2023) adopted the Fortran-C-CUDA C scheme and implemented a series of optimizations, including reduction the CPU–GPU communication frequency, optimize the GPU memory access, and thread and block co-indexing, to increase the computational efficiency of the HADVPPM advection solver. It can achieve up to the 18.8x speedup on the NVIDIA Tesla V100 GPU compared to the Intel Xeon Platinum 8168 CPU.*

**Line 81:**

Highlight

**Response:** Sorry for this mistake. We have revised this part in **lines 73-74**, which are as follows:

**Lines 73-74:** *The CUDA technology was implemented to carry out heterogeneous porting for the atmospheric chemical models from the CPU processors to different NVIDIA GPU accelerators.*

**Line 85:**

The Fortran

**Response:** Sorry for this mistake. We have revised this part in **lines 79-82** which are as follows:

**Lines 79-82:** *First, we compared the simulation result of the Fortran version CAMx model with the CUDA version of CAMx (CAMx-CUDA) and CAMx‐HIP model which were coupled with the CUDA and HIP versions of GPU-HADVPPM program, respectively.*

**Line 85:**

The CUDA

**Response:** Sorry for this mistake. We have revised this part in **lines 79-82**, which are as follows:

**Lines 79-82:** *First, we compared the simulation results of the Fortran version CAMx model with the CAMx-CUDA and CAMx-HIP model which were coupled with the CUDA and HIP versions of GPU-HADVPPM program, respectively.*

**Line 84:**

why is this a GPU-like accelerator and not simply "accelerator" or simply a GPU?

**Response:** Sorry for not being able to explain it clearly. In the previous study, we implement the NVIDIA GPU to accelerate the advection module in the CAMx model (Cao et al.,2023). In this study, the domestic GPU-like accelerator plays the same role as the NVIDIA GPU, which is also used to accelerate the advection module. As mentioned above, Chinese chip manufactures are reluctant to disclose the specific details of the GPU, probably due to the US ban on selling advanced GPUs to China, so we don't know the detailed architecture and can only refer to it as a GPU-like accelerator. We have added this part in **lines 74-79**, which are as follows:

**Lines 74-79:** *In this study, the Heterogeneous-computing Interface for Portability (HIP) interface was introduced to implement the porting of GPU-HADVPPM from the NVIDIA GPU to the China's domestically GPU-like accelerators based on the research of Cao et al. (2023). The domestic GPU-like accelerator plays the same role as the NVIDIA GPU, which is also used to accelerate the advection module in the CAMx model, so we refer to it as a GPU-like accelerator.*

**Line 86:**

the CUDA and HIP versions

**Response:** Sorry for this mistake. We have revised this part in **lines 79-82**, which are as follows:

**Lines 79-82:** *First, we compared the simulation results of the Fortran version CAMx model with the CAMx-CUDA and CAMx-HIP model which were coupled with the CUDA and HIP versions of GPU-HADVPPM program, respectively.*

**Line 88:**

Were

**Response:** Sorry for this mistake. We have revised this part in **lines 82-83**, which are as follows:

**Lines 82-83:** *And then, the computing performance of GPU-HADVPPM programs on different GPUs were compared.*

**Line 89:**

does this system have a name?

**Response:** Thanks for the constructive comment. In this paper, domestic heterogeneous cluster A refers to the "Songshan" supercomputer from the Henan Province, and domestic heterogeneous cluster B refers to the Taiyuan computing platform from the Shanxi Province. We have changed the names of all heterogeneous clusters accordingly.

**Line 105:**

this statement warrants a supporting reference

**Response:** Thanks for the constructive comment. We have added a supporting reference **lines 99-101**, which are as follows:

**Lines 99-101:** *The PPM horizontal advection scheme (HADVPPM) was selected in this study because it provides higher accuracy with minimal numerical diffusion (ENVIRON, 2014).*

**Line 106:**

Schemes

**Response:** Sorry for this mistake. We have revised this part in **lines 101-102**, which are as follows:

**Lines 101-102:** *The other numerical schemes selected during the CAMx model testing are listed in Table S1.*

**Line 106:**

do you maybe mean "testing" here?

**Response:** Sorry for not being able to explain it clearly. We have revised this part in **lines 105-106**, which are as follows:

**Lines 105-106:** *The other numerical schemes selected during the CAMx model testing are listed in Table S1.*

**Line 116:**

Strikeout

**Response:** Sorry for this mistake. We have revised this part in **lines 109-110**, which are as follows:

**Lines 109-110**: *CUDA is a proprietary application programming interface (API) and as such is only supported on NVIDIA's GPUs.*

**Line 123:**

Allowing

**Response:** Sorry for this mistake. We have revised this part in **lines 117-119**, which are as follows:

**Lines 117-119**: *On the ROCm software platform, it uses the AMD's HIP interface which is a C++*

*runtime API allowing developers to run programs on AMD GPUs.*

**Line 124:**

illustrates, or shows examples of the differences?

**Response:** Sorry for not being able to explain it clearly. Table 1 shows examples of the differences between the CUDA programming and HIP programming on the NVIDIA GPU and AMD GPU. Reviewer #1 gives the suggestion that Table 1 is unnecessary and need not be reproduced here. It is sufficient to refer to the HIP API. Therefore, we have removed Table 1.

**Line 126:**

they are very similar

**Response:** Sorry for this mistake. We have revised this part in **lines 119-120**, which are as follows:

**Lines 119-120:** *In general, they are very similar and their code can be converted directly by replacing the string "cuda" with "hip" in the most cases.*

**Line 126:**

String

**Response:** Sorry for this mistake. We have revised this part in **lines 119-120**, which are as follows:

**Lines 119-120:** *In general, they are very similar and their code can be converted directly by replacing the string "cuda" with "hip" in the most cases.*

**Line 126:**

Is

**Response:** Sorry for this mistake. We have revised this part in **lines 120-121**, which are as follows:

**Lines 120-121:** *More information about HIP API is available on the AMD ROCm website (ROCm, 2023).*

**Line 130:**

does this GPU-like accelerator have a name?

**Response:** Sorry for not being able to explain it clearly. As mentioned above, Chinese chip manufactures are reluctant to disclose the specific details of the GPU, probably due to the US ban

on selling advanced GPUs to China, so we don't know the name of this GPU-like accelerator either.

**Line 129:**

Presumably it is not possible to run CUDA code on the Chinese accelerators. Please state so explicitly.

As I'm sure it is clear for the authors, the lack of details of the accelerators implies readers are unaware of the software required to program this hardware.

**Response:** Sorry for not being able to explain it clearly. The CUDA code cannot run directly on domestic GPU-like accelerators, and it needs to be transcoded into HIP code. As mentioned above, Chinese chip manufactures are reluctant to disclose the specific details of the GPU, probably due to the US ban on selling advanced GPUs to China, so we don't more specific information, and all the information we know as well as are listed in Table 2. We have revised this part in **lines 121-124**, which are as follows:

**Lines 121-124:** *Similar to AMD GPU, developers can also use ROCM-HIP programming interface to implement programs running on the China's domestically GPU-like accelerator. The CUDA code cannot run directly on domestic GPU-like accelerators, and it needs to be transcoded into HIP code.*

**Line 135:**

Lists

**Response:** Sorry for this mistake. We have revised this part in **lines 126-129**, which are as follows:

**Lines 126-129:** *Table 1 lists four GPU clusters where we conducted the experiments, two NVIDIA heterogeneous clusters which have the same hardware configuration as Cao et al. (2023) and two China's domestically heterogeneous clusters newly used in this research, namely "Songshan" supercomputer and "Taiyuan" computing platform.*

**Line 135:**

Where we

**Response:** Sorry for this mistake. We have revised this part in **lines 126-129**, which are as follows:

**Lines 126-129:** *Table 1 lists four GPU clusters where we conducted the experiments, two NVIDIA heterogeneous clusters which have the same hardware configuration as Cao et al. (2023) and two China's domestically heterogeneous clusters newly used in this research, namely "Songshan" supercomputer and "Taiyuan" computing platform.*

**Line 147:**

Units

**Response:** Sorry for this mistake. Reviewer #1 gave a suggestion that most of the technical information was repeated in Table 1 and could be omitted from the text, so we removed some of the contents of lines 126-144 that were duplicated with Table 1.

**Line 147:**

not sure "totaling" means here... does it mean each of the 64 units can compute up to 60 parallel threads?

**Response:** Sorry for not being able to explain it clearly. The GPU-like accelerator A has 64 compute units, and each of the compute units can compute up to 60 parallel threads. We have revised this part in Table 1.

**Line 147:**

In general, providing names for the clusters, the CPUs and the accelerators would improve readability and would be more informative for the readers.

Some references to additional technical specifications of the CPUs and accelerators would also be very welcome.

**Response:** Sorry for not being able to explain it clearly. In this paper, domestic heterogeneous cluster A refers to the "Songshan" supercomputer from the Henan Province, and domestic heterogeneous cluster B refers to the Taiyuan computing platform from the Shanxi Province. We have changed the names of all heterogeneous clusters accordingly. As mentioned above, Chinese chip manufactures are reluctant to disclose the specific details of the GPU, probably due to the US ban on selling advanced GPUs to China, so we don't more specific information, and all the information we know as well as are listed in Table 1.

*Table 1. Configurations of the NVIDIA K40m cluster, NVIDIA V100 cluster, "Songshan" supercomputer, and "Taiyuan" computing platform.*

| | Hardware components | |
| --- | --- | --- |
| | *CPU* | *GPU* |
| *NVIDIA K40m cluster* | *Intel Xeon E5-2682 v4 CPU @2.5 GHz, 16 cores* | *NVIDIA Tesla K40m GPU, 2880 CUDA cores, 12 GB video memory* |
| *NVIDIA V100 cluster* | *Intel Xeon Platinum 8168 CPU @2.7 GHz, 24 cores* | *NVIDIA Tesla V100 GPU, 5120 CUDA cores, 16 GB video memory* |
| *Songshan supercomputer* | *China's domestically CPU processor A, 2.0GHz, 32 cores* | *China's domestically GPU-like accelerator A, 3840 computing units, 16 GB memory* |

| | China's domestically CPU processor B, 2.5GHz, 32 cores | China's domestically GPU-like accelerator B, 8192 computing units, 16 GB memory |
|---|---|---|
| *Taiyuan computing platform* | | |
| | *Software environment* | |
| | *Compiler and MPI* | *Programming model* |
| *NVIDIA K40m cluster* | *Intel Toolkit 2021.4.0* | *CUDA-10.2* |
| *NVIDIA V100 cluster* | *Intel Toolkit 2019.1.144* | *CUDA-10.0* |
| *Songshan supercomputer* | *Intel Toolkit 2021.3.0* | *ROCm-4.0.1/ DTK-23.04* |
| *Taiyuan computing platform* | *Intel Toolkit 2021.3.0* | *DTK-23.04* |

**Line 155:**

In terms

**Response:** Sorry for this mistake. We have revised this part in **lines 141-143**, which are as follows:

**Lines 141-143:** *In terms of the software environment, the NVIDA GPU is programmed using the CUDA toolkit, and the domestic GPU-like is programmed using the ROCm-HIP toolkit developed by AMD (ROCm, 2023).*

**Line 173:**

Strikeout

**Response:** Sorry for this mistake. We have revised this part in **lines 155-157**, which are as follows:

**Lines 155-157:** *Then the CUDA and ROCm HIP technology were used to convert the standard C code into CUDA C and HIP C code to make it computable on the NIVIDA GPU and domestic GPU-like accelerator.*

**Line 176:**

Strikeout

**Response:** Sorry for this mistake. We have revised this part in **lines 166-168**, which are as follows:

**Lines 166-168:** *In this study, the ROCm HIP technology was used to implement the operation of GPU-HADVPPM on domestic GPU-like accelerator based on the CUDA version of GPU-HADVPPM which was developed by Cao et al. (2023).*

**Line 179:**

Strikeout

**Response:** Sorry for this mistake. We have revised this part in **lines 168-169**, which are as follows:

**Lines 168-169:** *The HIP code was compiled using the "hipcc" compiler driver with the library flag "-lamdhip64".*

**Line 192:**

what exactly does this mean?

**Response:** Sorry for not being able to explain it clearly. We have revised this part in **lines 179-182**, which are as follows:

**Lines 179-182:** *During the process of CAMx model simulation, MPI and OMP hybrid parallelism can be used, several CPU processes can be launched, and each process can spawn several threads. This hybrid parallelism can significantly improve the computational efficiency of CAMx model.*

**Line 193:**

Strikeout

**Response:** Sorry for this mistake. We have revised this part in **lines 187-189**, which are as follows:

**Lines 187-189:** *In the previous studying, Cao et al. (2023) adopt a parallel architecture with an MPI and CUDA (MPI+CUDA) hybrid paradigm to configure one GPU accelerator for each CPU process.*

**Line 208:**

Are reported

**Response:** Sorry for this mistake. We have revised this part in **lines 206-207**, which are as follows:

**Lines 206-207:** *The computational performance experiments of CUDA and HIP version GPU-HADVPPM are reported in this section.*

**Line 208:**

The

**Response:** Sorry for this mistake. We have revised this part in **lines 207-209**, which are as follows:

**Lines 207-209:** *First, we compared the simulation result of the Fortran version CAMx model with CAMx-CUDA and CAMx-HIP model which were coupled with CUDA and HIP version of GPU-HADVPPM program, respectively.*

**Lines 203-205:**

perhaps it would be useful to provide a sketch of how the different types of processes map to hardware, both with and without the OpenMP for the "other modules".

**Response:** Sorry for not being able to explain it clearly and thanks for the constructive comment. The original CAMx model supports message passing interface (MPI) parallel technology running on the general-purpose CPU. The simulation domain is divided into several sub-regions by MPI, and each CPU process is responsible for simulation of its sub-region, which includes advection module and other modules such as photolysis module, deposition module, chemical module, etc. In the previous studying, Cao et al. (2023) adopt a parallel architecture with an MPI and CUDA (MPI+CUDA) hybrid paradigm to configure one GPU accelerator for each CPU process. For the advection module, the simulation originally implemented by the CPU is handed over to the GPU. Other module computing tasks continue to be completed on the CPU. In this study, in addition to the MPI+HIP hybrid parallel scheme to realize the advection module running on several domestic GPU-like accelerators, the OpenMP (OMP) hybrid parallel framework of CAMx model is further introduced to realize the MPI+OMP hybrid parallelism of other modules on CPU. A schematic of the multi-level hybrid parallel framework is shown in Figure 2. For example, in a computing node, four CPU processes and four GPU-like accelerators are launched, and each CPU process spawns four threads. Then the advection module is simulated by 4 GPU-like accelerators, and the other modules are done by 4*4 threads spawned by CPU processes. We have modified this part in **lines 179-191**, which are as follows:

**Lines 179-191:** *As mentioned above, the original CAMx model supports message passing interface (MPI) parallel technology running on the general-purpose CPU. The simulation domain is divided into several sub-regions by MPI, and each CPU process is responsible for computation of its sub-region, which includes the computation tasks of advection module and other modules such as photolysis module, deposition module, chemical module, etc. In the previous studying, Cao et al. (2023) adopt a parallel architecture with an MPI and CUDA (MPI+CUDA) hybrid paradigm to configure one GPU accelerator for each CPU process. For the advection module, the simulation originally implemented by the CPU is handed over to the GPU. Other module computing tasks continue to be completed on the CPU.*

*In this study, when the CUDA C code of GPU-HADVPPM is converted to HIP C code, GPU-HADVPPM with an MPI and HIP (MPI+HIP) heterogeneous hybrid programming technology can also run on multiple domestic GPU-like accelerators. However, the number of GPU-like accelerators in a single compute node is usually much smaller than the number of CPU cores in the heterogeneous HPC systems. Therefore, in order to make full use of the remaining CPU computing resources, the OMP API of CAMx model is further introduced to realize the MPI+OMP hybrid parallelism of other modules on CPU. A schematic of the multi-level hybrid parallel framework is*

*shown in Figure 2. For example, in a computing node, four CPU processes and four GPU-like accelerators are launched, and each CPU process spawns four threads. Then the advection module is simulated by 4 GPU-like accelerators, and the other modules are done by 4\*4 threads spawned by CPU processes.*

[Figure]

**Figure 2.** *A schematic of the multi-level hybrid parallel framework.*

**Lines 216-224:**

presumably these test cases have been used already in previous studies. Please refer to them for completeness.

**Response:** Sorry for not being able to explain it clearly. In the previous study of Cao et al. (2023), we only used BJ case to carry out the performance test, HN case and ZY case are the newly constructed test cases in this study, so we think it is necessary to introduce the experimental setup of the three test cases in detail. We have revised this part in **lines 216-226**, which are as follows:

**Lines 216-226:** *There are three test cases were used to evaluate the performance of CUDA and HIP version GPU-HADVPPM. The experimental setup for the three test cases is shown in Table 2. In the previous study of Cao et al. (2023), the BJ case was used to carry out the performance tests, HN case and ZY case are the newly constructed test cases in this study. The Beijing case (BJ) covers Beijing, Tianjin, and part of the Hebei Province with 145 × 157 grid boxes, and simulation of BJ case starts on 1 November, 2020. The Henan case (HN) mainly covers the Henan Province with 209 × 209 grid boxes. The starting date of simulation in HN case is 1 October, 2022. The Zhongyuan*

*case (ZY) has the widest coverage of the three cases, with Henan Province as the center, covering the Beijing-Tianjin-Hebei region, Shanxi Province, Shaanxi Province, Hubei Province, Anhui Province, Jiangsu Province, and Shandong Province, with 531 × 513 grid boxes. ZY case started simulation on 4 January, 2023.*

**Line 236:**

By the

**Response:** Sorry for this mistake. We have revised this part in **lines 236-238**, which are as follows:

**Lines 236-238:** *The hourly concentrations of four major species, i.e. $O_3$, $PSO_4$, $CO$, and $NO_2$, outputted by the Fortran, CUDA, and HIP versions of CAMx for the BJ case are compared to verify the results correctness before testing the computational performance.*

**Line 236:**

Versions

**Response:** Sorry for this mistake. We have revised this part in **lines 236-238**, which are as follows:

**Lines 236-238:** *The hourly concentrations of four major species, i.e. $O_3$, $PSO_4$, $CO$, and $NO_2$, outputted by the Fortran, CUDA, and HIP versions of CAMx for the BJ case are compared to verify the results correctness before testing the computational performance.*

**Line 237:**

Correctness

**Response:** Sorry for this mistake. We have revised this part in **lines 236-238**, which are as follows:

**Lines 236-238:** *The hourly concentrations of four major species, i.e. $O_3$, $PSO_4$, $CO$, and $NO_2$, outputted by the Fortran, CUDA, and HIP versions of CAMx for the BJ case are compared to verify the results correctness before testing the computational performance.*

**Line 237:**

Computational

**Response:** Sorry for this mistake. We have revised this part in **lines 236-238**, which are as follows:

**Lines 236-238:** *The hourly concentrations of four major species, i.e. $O_3$, $PSO_4$, $CO$, and $NO_2$, outputted by the Fortran, CUDA, and HIP versions of CAMx for the BJ case are compared to verify the results correctness before testing the computational performance.*

**Line 237:**

Shows

**Response:** Sorry for this mistake. We have revised this part in **lines 238-242**, which are as follows:

**Lines 238-242:** *Fig. 3 shows the four major species simulation results of the three CAMx version, including Fortran version on the Intel E5-2682 v4 CPU, CUDA version on the NVIDIA K40m cluster and HIP version on the "Songshan" supercomputer, after 48 hours integration, as well as the absolute errors (AEs) of their concentrations.*

**Line 238:**

The three

**Response:** Sorry for this mistake. We have revised this part in **lines 238-242**, which are as follows:

**Lines 238-242:** *Fig. 3 shows the four major species simulation results of the three CAMx version, including Fortran version on the Intel E5-2682 v4 CPU, CUDA version on the NVIDIA K40m cluster and HIP version on the "Songshan" supercomputer, after 48 hours integration, as well as the absolute errors (AEs) of their concentrations.*

**Line 240:**

relative to the serial Fortran version? Please state this explicitly.

**Response:** Sorry for not being able to explain it clearly. According to the results of Cao et al. (2023), the parallel design of the CAMx model adopts the primary/secondary mode, and Process 0 (P0) is responsible for inputting and outputting the data and calling the MPI_Barrier function to synchronize the process, and the other processes are responsible for simulation. When comparing the simulation results, we only launched 2 CPU processes on the CPU platform, and launched 2 CPU processes and configure 2 GPU accelerators on the NVIDIA K40m and "Songshan" supercomputer respectively. We have revised this part in **lines 242-247**, which are as follows:

**Lines 242-247:** *As described by Cao et al. (2023), the parallel design of the CAMx model adopts the primary/secondary mode, and P0 process is responsible for inputting and outputting the data and calling the MPI_Barrier function to synchronize the process, and the other processes are responsible for simulation. When comparing the simulation results, we only launched 2 CPU processes on the CPU platform, and launched 2 CPU processes and configure 2 GPU accelerators on the NVIDIA K40m cluster and "Songshan" supercomputer, respectively.*

**Line 250:**

Not being knowledgeable of how the solver is implemented, are the concentrations of the different species (in terms of ppbV and microgram/m3) the state variables solved? Or are they derived quantities?

The reason I ask is, assuming the algorithms are identical, the differences must then come from arithmetic precision on the different implementations. One would hope to see errors for the state variables in the order of arithmetic precision of the hardware.

Please elaborate and discuss this.

**Response:** Thanks for the constructive suggestion. Fig. 3 shows the four major species simulation results of the three CAMx version, including Fortran version on the Intel E5-2682 v4 CPU, CUDA version on the NVIDIA K40m cluster and HIP version on the "Songshan" supercomputer, after 48 hours integration, as well as the absolute errors (AEs) of their concentrations. In order to further explore the AEs accumulation and growth between the CUDA version and HIP version of CAMx model and the original Fortran version after heterogeneous porting, Figure 3 highlights the time series of AEs between Fortran and CUDA versions and between Fortran and HIP versions after grid averaging. As is shown in Figure 3, the AEs of $O_3$, $PSO_4$, CO, and $NO_2$ between the Fortran version and the CUDA version are -0.0002 to 0.0001 ppbV, -0.00003 to 0.00001 $\mu g \cdot m^{-3}$, -0.0004 to 0.0004 ppbV, and -0.0002 to 0.0002 ppbV, respectively, and fluctuate. Although the AEs of the above four species between the Fortran and the HIP version also fluctuates, the fluctuation range is much smaller than that of the CUDA version. Importantly, the AEs between Fortran and CUDA versions and between Fortran and HIP versions both do not accumulate and grow over prolonged simulation periods. We have added this discussion in **lines 259-265**, which are as follows:

**Lines 259-265:** *As is shown in Fig. 4, the AEs of $O_3$, $PSO_4$, CO, and $NO_2$ between the Fortran version and the CUDA version are -0.0002 to 0.0001 ppbV, -0.00003 to 0.00001 $\mu g \cdot m^{-3}$, -0.0004 to 0.0004 ppbV, and -0.0002 to 0.0002 ppbV, respectively, and fluctuate. Although the AEs of the above four species between the Fortran and the HIP version also fluctuates, the fluctuation range is much smaller than that of the CUDA version. Importantly, the AEs between Fortran and CUDA versions and between Fortran and HIP versions both do not accumulate and grow over prolonged simulation periods.*

[Figure]

**Figure 4.** *The time series of AEs between Fortran and CUDA versions (solid blue line) and between Fortran and HIP versions (solid red line) after grid averaging. Panel (a)~(d) represent the AEs of $O_3$, $PSO_4$, $CO$, and $NO_2$, respectively.*

**Line 265 Figure 3:**

Is the far larger range of errors for PS04 explainable?

More generally, if the errors come from hardware-dependent arithmetic, how does this transfer to different ranges of error for the different species? What explains this?

**Response:** Sorry for not being able to explain it clearly. $PSO_4$ is selected as the representative of secondary particulate matter. In the air quality model, the secondary particulate matter, such as $PNH_4$, $PNO_3$, and $PSO_4$, have a common characteristic: their initial concentration is very low and they are mainly generated through complex chemical reactions. Therefore, when calculating the relative error on different hardware platforms, because the value in the denominator is very small, it is very sensitive to a small difference in the numerator, resulting in a large relative error. But from the absolute error in Fig.1, the absolute error of $PSO_4$ on different hardware platforms is smaller than that of other species. Fig. 3 presents the boxplot of the relative errors (REs) in all grid boxes for the $PSO_4$, $PNO_3$, $PNH_4$, $O_3$, $CO$, and $NO_2$ during the 48 hours simulation under the BJ case. As mentioned above, the initial concentration of secondary particles such as $PSO_4$, $PNO_3$ and $PNH_4$ is very small. Therefore, when calculating the relative error, a small difference in the numerator leads to a large relative error value. For gaseous pollutants such as $CO$, $O_3$, and $NO_2$, the initial concentration is large due to emission, and the denominator value is large when calculating the relative error, which is insensitive to small differences in the numerator. We have revised this part in **lines 277-293**, which are as follows:

**Lines 277-293:** *Fig. 5 presents the boxplot of the relative errors (REs) in all grid boxes for the PSO4, PNO3, PNH4, O3, CO, and NO2 during the 48 hours simulation under the BJ case. Statistically,*

*the REs between the CUDA version on the NVIDIA K40m cluster and Fortran version on the Intel E5-2682 v4 CPU for the above six species are in the range of ±0.006%, ±0.01%, ±0.008%, ±0.002%, ±0.002%, and ±0.002%. In terms of REs between the HIP version on the "Songshan" supercomputer and Fortran version on the Intel E5-2682 v4 CPU, the values are much smaller than REs between CUDA and Fortran versions which are fall into the range of ±0.0005%, ±0.004%, ±0.004%, ±0.00006%, ±0.00004%, and ±0.00008%, respectively. In the air quality model, the secondary particulate matter, such as PNH4, PNO3, and PSO4, have a common characteristic: their initial concentration is very low and they are mainly generated through complex chemical reactions. Therefore, when calculating the relative error on different hardware platforms, because the value in the denominator is very small, it is very sensitive to a small difference in the numerator, resulting in a large relative error. But from the absolute error in Fig.3, the absolute error of PSO4 on different hardware platforms is smaller than that of other species. For gaseous pollutants such as CO, O3, and NO2, the initial concentration is large due to emission, and the denominator value is large when calculating the relative error, which is insensitive to small differences in the numerator.*

[Figure]

***Figure 5.*** *The distribution of REs in all grid boxes for the PSO₄, PNO₃, PNH₄, O₃, CO, and NO₂ under the BJ case. The red boxplot represents the REs between the CUDA version on the NVIDIA K40m cluster and Fortran version on the Intel E5-2682 v4 CPU, and blue boxplot represents the REs between the HIP version on the "Songshan" supercomputer and Fortran version on the Intel E5-2682 v4 CPU.*

**Line 288:**

how can you support this claim? can you show this with simple kernels, or provide references for this?

**Response:** Sorry for not being able to explain it clearly. In this paper, by comparing the simulation results of different versions of CAMx, it can be seen that GPU-HADVPPM4HIP has less deviation

on domestic GPU-like accelerators. Because the simulation accuracy of geoscience numerical model is closely related to the model efficiency, and many model optimization works improve the computational performance by reducing the precision of the data, such as Váňa et al. (2017) changed some variables precision in the atmospheric model from double precision to single precision, which increased the overall computational efficiency by 40%, and Wang et al. (2019) improved the computational efficiency of the gas-phase chemistry module in the air quality mode by 25%~28% by modifying the floating-point precision compile flag. Therefore, we speculate that this may be related to the manufacturing process of NVIDIA GPUs and domestic GPU-like accelerators, especially NIVIDA Tesla V100 series GPUs, which may use unknown optimizations to improve GPU performance efficiency by losing part of the accuracy. In this study, we mainly focus on numerical simulation, and the difference of hardware is not in the scope of our study. Of course, we also want to know the specific reasons for this, but we are not professional GPU research and development designers after all and do not know the underlying design logic of the hardware, so we can only present our experimental results in the air pollution model to you, and discuss with each other to jointly promote the application of GPU in the field of geoscience numerical models. We have added this discussion in **lines 317-333**, which are as follows:

**Lines 317-333:** *From AEs, REs, and ratio of RMSE and std between different CAMx versions, it is less difference that the GPU-HADVPPM4HIP program runs on the "Songshan" supercomputer. Because the simulation accuracy of geoscience numerical model is closely related to the model efficiency, and many model optimization works improve the computational performance by reducing the precision of the data, such as Váňa et al. (2017) changed some variables precision in the atmospheric model from double precision to single precision, which increased the overall computational efficiency by 40%, and Wang et al. (2019) improved the computational efficiency of the gas-phase chemistry module in the air quality mode by 25%~28% by modifying the floating-point precision compile flag. Therefore, we speculate that this may be related to the manufacturing process of NVIDIA GPUs and domestic GPU-like accelerators, especially NIVIDA Tesla V100 series GPUs, which may use unknown optimizations to improve GPU performance efficiency by losing part of the accuracy. In this study, we mainly focus on numerical simulation. Of course, we also want to know the specific reasons for this, but we are not professional GPU research and development designers after all and do not know the underlying design logic of the hardware, so we can only present our experimental results in the air pollution model to you, and discuss with each other to jointly promote the application of GPU in the field of geoscience numerical models.*

**Line 296:**

I'm not sure what you meant here

**Response:** Sorry for not being able to explain it clearly. We have revised this part in **lines 336-337**, which are as follows:

**Lines 336-337:** *As described in Sect. 4.2, we validate the 48 hours simulation results outputted by the Fortran, CUDA, and HIP versions of CAMx.*

**Lines 297-301:**

the syntax here is wrong.

I think you mean something like this:

Next, the computational performance were compared for the Fortran version...., the CUDA version. .... and the HIP version...., under the BJ, HN and ZY case.

It is unclear to me here what "coupling" is intended to mean.

**Response:** Sorry for not being able to explain it clearly. We have revised this part in **lines 337-341**, which are as follows:

**Lines 337-341:** *Next, computational performance was compared for the Fortran version of HADVPPM on the Intel Xeon E5-2682 v4 CPU and domestic CPU processor A, the CUDA version of GPU-HADVPPM on the NVIDIA Tesla K40m and V100 GPU, and the HIP version of GPU-HADVPPM on the domestic GPU-like accelerator A, under the BJ, HN and ZY case.*

**Line 302:**

Lists

**Response:** Sorry for this mistake. We give the relevant data in Table 5 as a picture, and change the word "list" to "show". This part has been revised in **lines 302-303**, which are as follows:

**Lines 302-303:** *Fig. 6(a) and (b) show the elapsed time and speedup of the different versions of HADVPPM on the CPU processors and GPU accelerators for BJ, HN, and ZY cases, respectively.*

**Lines 318-321:**

table 5 might be more informative for the reader in the form of a bar chart.

If I understand this correctly, speed up is computed relative to the elapsed time of the domestic CPU processor A times. Please state this explicitly. Please also state explicitly whether the runs on the CPU (both with the Chinese CPU and the Intel CPUs) are using OpenMP or serial.

**Response:** Thanks for the constructive suggestion. We have converted Table 5 into bar and line charts, and the specific form is shown in Figure 5. When evaluating the computational efficiency on different hardware platforms, the elapsed time of advection module launched two CPU processes on the domestic CPU processor A is taken as the benchmark, that is, the speedup is 1.0x. The runtime of the advection module on Intel CPU processor and different GPU accelerators is compared with

the baseline to obtain the speedup. As mentioned above, when two CPU processes are launched to run CAMx, only the P1 process is involved in the computing, and the P0 process is mainly responsible for data input/output and communication. We have revised this part in **lines 336-366**, which are as follows:

**Lines 336-366:** *As described in Sect. 4.2, we validate the 48 hours simulation results outputted by the Fortran, CUDA, and HIP versions of CAMx. Next, computational performance was compared for the Fortran version of HADVPPM on the Intel Xeon E5-2682 v4 CPU and domestic CPU processor A, the CUDA version of GPU-HADVPPM on the NVIDIA Tesla K40m and V100 GPU, and the HIP version of GPU-HADVPPM on the domestic GPU-like accelerator A, under the BJ, HN and ZY case. The simulation time in this section is 1 hour unless otherwise specified.*

*Similarity, since the CAMx model adopts the primary/secondary mode, two CPU processes P0 and P1 are launched on the CPU, and the system_clock functions in the Fortran language are used to test the elapsed time of the advection module in P1 process. When testing the computation performance of the advection module on the GPU-like accelerator, we also only launch 2 CPU processes and 2 GPU-like accelerators. When a P1 process runs to the advection module, the original computation process is migrated from the CPU to the GPU, and the hipEvent_t function in HIP programming is used to test the running time of the advection module on the GPU-like accelerator. When comparing the speedup on different GPU accelerators, the elapsed time of advection module launched one CPU process (P1) on the domestic CPU processor A is taken as the benchmark, that is, the speedup is 1.0x. The runtime of the advection module on Intel CPU processor and different GPU accelerators is compared with the baseline to obtain the speedup.*

*Fig. 6(a) and (b) shows the elapsed time and speedup of the different versions of HADVPPM on the CPU processors and GPU accelerators for BJ, HN, and ZY cases, respectively. The results show that using CUDA and HIP technology to port HADVPPM from CPU to GPU can significantly improve its computational efficiency. For example, the elapsed time of the advection module on the domestic processor A is 609.2 seconds under the ZY case. After it is ported to the domestic GPU accelerator and NVIDIA V100 GPU, it only takes 21.1 seconds and 7.6 seconds to complete the computing, and the speedups are 28.9x and 80.2x, respectively. The ZY case had the largest number of grids in the three cases and exceeded the memory of a single NVIDIA Tesla K40m GPU accelerator, so it was not possible to test its elapsed time on it. Moreover, the optimization of thread and block co-indexing is used to simultaneously compute the grid point in the horizontal direction (Cao et al., 2023). Therefore, it can be seen from Fig. 6(b) that the larger the computing scale, the more obvious the acceleration, which indicates that GPU is more suitable for super-large scale parallel computing, and provides technical support for accurate and fast simulation of ultra-high-resolution air quality at the meter level in the future.*

[Figure]

**Figure 6.** *The elapsed time (a) and speedup (b) of the Fortran version of HADVPPM on the Intel Xeon E5-2682 v4 CPU and the domestic CPU processor A, the CUDA version of GPU-HADVPPM on the NVIDIA Tesla K40m GPU, NVIDIA Tesla V100 GPU, and the HIP version of GPU-HADVPPM on the domestic GPU-like accelerator A for BJ, HN, and ZY case. The unit of elapsed time is in seconds (s).*

**Lines 307-313:**

I think this information is redundant to Table 5. I would suggest to remove it from the text.

It seems the point you intend to highlight is that the speed up is larger for the larger case. I would suggest to write a sentence specifically on that. The readers can check the numbers in the table.

**Response:** Thanks for the constructive suggestion. We have removed the redundant information from the text, and revised this part in **lines 353-366**, which are as follows:

**Lines 353-366:** *Fig. 6(a) and (b) shows the elapsed time and speedup of the different versions of HADVPPM on the CPU processors and GPU accelerators for BJ, HN, and ZY cases, respectively. The results show that using CUDA and HIP technology to port HADVPPM from CPU to GPU can significantly improve its computational efficiency. For example, the elapsed time of the advection*

*module on the domestic processor A is 609.2 seconds under the ZY case. After it is ported to the domestic GPU accelerator and NVIDIA V100 GPU, it only takes 21.1 seconds and 7.6 seconds to complete the computing, and the speedups are 28.9x and 80.2x, respectively. The ZY case had the largest number of grids in the three cases and exceeded the memory of a single NVIDIA Tesla K40m GPU accelerator, so it was not possible to test its elapsed time on it. Moreover, the optimization of thread and block co-indexing is used to simultaneously compute the grid point in the horizontal direction (Cao et al., 2023). Therefore, it can be seen from Fig. 6(b) that the larger the computing scale, the more obvious the acceleration, which indicates that GPU is more suitable for super-large scale parallel computing, and provides technical support for accurate and fast simulation of ultra-high-resolution air quality at the meter level in the future.*

**Line 326:**

Missing (

**Response:** Sorry for this mistake. We have revised this part in **lines 394-397**, which are as follows:

**Lines 394-397:** *However, the communication bandwidth of data transfer between the CPU and GPU is one of the most significant factors that restrict the performance of numerical model on the heterogeneous cluster (Mielikainen et al., 2012; Mielikainen et al., 2013; Huang et al., 2013).*

**Line 327:**

study, explore, illustrate?

**Response:** Sorry for not being able to explain it clearly. We have revised this part in **lines 397-401**, which are as follows:

**Lines 397-401:** *To illustrate the significant impact of CPU-GPU data transfer efficiency, the computational performance of GPU-HADVPPM with and without data transfer time for the BJ case is tested on the "Songshan" supercomputer and "Taiyuan" computing platform with the same DTK version 23.04 software environment and the results are further presented in Table 6.*

**Line 322:**

Coupled?

**Response:** Sorry for not being able to explain it clearly. We have revised this part in **lines 393-394**, which are as follows:

**Lines 393-394:** *The elapsed time of GPU-HADVPPM given in Table 4 on NVIDIA GPU and domestic GPU-like accelerator does not consider the data transfer time between CPU and GPU.*

**Line 327:**

Coupled?

**Response:** Sorry for not being able to explain it clearly. We have revised this part in **lines 397-401**, which are as follows:

**Lines 397-401:** *To illustrate the significant impact of CPU-GPU data transfer efficiency, the computational performance of GPU-HADVPPM with and without data transfer time for the BJ case is tested on the "Songshan" supercomputer and "Taiyuan" computing platform with the same DTK version 23.04 software environment and the results are further presented in Table 6.*

**Lines 329-331:**

this means the time that the accelerators are actually computing? Please call it something like kernel execution time or something along these lines. The other time is of course simply the total elapsed time or total runtime.

**Response:** Sorry for not being able to explain it clearly. We refer to the execution time of GPU-HADVPPM program on GPU kernel as kernel execution time, and the time of GPU-HADVPPM running on GPU as total runtime, which contains two parts, namely, kernel execution time and data transfer time between CPU-GPU. We have revised this part in **lines 401-408**, which are as follows:

**Lines 401-408:** *For convenience of description, we refer to the execution time of GPU-HADVPPM program on GPU kernel as kernel execution time, and the time of GPU-HADVPPM running on GPU as total runtime, which contains two parts, namely, kernel execution time and data transfer time between CPU and GPU. After testing, the kernel execution time and total running time of GPU-HADVPPM4HIP program on domestic GPU-like accelerator A are 6.8 and 29.8 seconds, respectively. In other words, it only takes 6.8 seconds to complete the computation on the domestic accelerator, but it takes 23.0 seconds to complete the data transfer between the CPU and the domestic GPU-like accelerator, which is 3.4 times the computation time.*

**Lines 332-333:**

This is really very inefficient.

**Response:** By comparing the kernel execution time and total running time of GPU-HADVPPM4HIP on the domestic accelerator, it can be seen that the data transfer efficiency between CPU-GPU is really inefficient, which seriously restricts the computational performance of numerical models in heterogeneous clusters. On the one hand, improving the data transfer bandwidth between CPU-GPU can improve the computational efficiency of the model in heterogeneous clusters. On the other hand, the optimization measures can be implemented to improve the data transfer efficiency between CPU-GPU. For example, (1) Asynchronous data

transfer is used to reduce the communication latency between CPU and GPU. Computation and data transfer are performed simultaneously to hide communication overhead; (2) Currently, some advanced GPU architectures support a unified memory architecture, so that the CPU and GPU can share the same memory space and avoid frequent data transfers. This reduces the overhead of data transfer and improves data transfer efficiency; (3) Cao et al. (2023) adopted communication optimization measures to reduce the communication frequency in one time integration step to one, but there is still the problem of high communication frequency in the whole simulation. In the future, we will consider porting other hotspots of CAMx model, or even the whole integral module except I/O, to GPU-like accelerators for increasing the proportion of code on the GPU and reduce the frequency of CPU-GPU communication. We have revised this part in **lines 412-428**, which are as follows:

**Lines 412-428:** *By comparing the kernel execution time and total running time of GPU-HADVPPM4HIP on the domestic accelerator, it can be seen that the data transfer efficiency between CPU and GPU is really inefficient, which seriously restricts the computational performance of numerical models in heterogeneous clusters. On the one hand, improving the data transfer bandwidth between CPU and GPU can improve the computational efficiency of the model in heterogeneous clusters. On the other hand, the optimization measures can be implemented to improve the data transfer efficiency between CPU and GPU. For example, (1) Asynchronous data transfer is used to reduce the communication latency between CPU and GPU. Computation and data transfer are performed simultaneously to hide communication overhead; (2) Currently, some advanced GPU architectures support a unified memory architecture, so that the CPU and GPU can share the same memory space and avoid frequent data transfers. This reduces the overhead of data transfer and improves data transfer efficiency; (3) Cao et al. (2023) adopted communication optimization measures to reduce the communication frequency in one time integration step to one, but there is still the problem of high communication frequency in the whole simulation. In the future, we will consider porting other hotspots of CAMx model, or even the whole integral module except I/O, to GPU-like accelerators for increasing the proportion of code on the GPU and reduce the frequency of CPU-GPU communication.*

**Lines 335-337:**

how do you propose to do this?

**Response:** For the problem of inefficiency of data transfer between CPU-GPU, we plan to carry out performance optimization in the following three aspects in the future. (1) Asynchronous data transfer is used to reduce the communication latency between CPU and GPU. Computation and data transfer are performed simultaneously to hide communication overhead; (2) Currently, some advanced GPU architectures support a unified memory architecture, so that the CPU and GPU can share the same memory space and avoid frequent data transfers. This reduces the overhead of data

transfer and improves data transfer efficiency; (3) Cao et al. (2023) adopted communication optimization measures to reduce the communication frequency in one time integration step to one, but there is still the problem of high communication frequency in the whole simulation. In the future, we will consider porting other hotspots of CAMx model, or even the whole integral module except I/O, to GPU-like accelerators for increasing the proportion of code on the GPU and reduce the frequency of CPU-GPU communication. We have revised this part in **lines 412-428**, which are as follows:

**Lines 412-428:** *By comparing the kernel execution time and total running time of GPU-HADVPPM4HIP on the domestic accelerator, it can be seen that the data transfer efficiency between CPU and GPU is really inefficient, which seriously restricts the computational performance of numerical models in heterogeneous clusters. On the one hand, improving the data transfer bandwidth between CPU and GPU can improve the computational efficiency of the model in heterogeneous clusters. On the other hand, the optimization measures can be implemented to improve the data transfer efficiency between CPU and GPU. For example, (1) Asynchronous data transfer is used to reduce the communication latency between CPU and GPU. Computation and data transfer are performed simultaneously to hide communication overhead; (2) Currently, some advanced GPU architectures support a unified memory architecture, so that the CPU and GPU can share the same memory space and avoid frequent data transfers. This reduces the overhead of data transfer and improves data transfer efficiency; (3) Cao et al. (2023) adopted communication optimization measures to reduce the communication frequency in one time integration step to one, but there is still the problem of high communication frequency in the whole simulation. In the future, we will consider porting other hotspots of CAMx model, or even the whole integral module except I/O, to GPU-like accelerators for increasing the proportion of code on the GPU and reduce the frequency of CPU-GPU communication.*

**Line 333:**

this seems to imply that you attribute the data transfer issues strictly on the hardware. Implementation, both on the software stack and in the solver itself can play a very large role. Why do you go directly to hardware?

**Response:** Sorry for not being able to explain it clearly. Data transfer between CPU-GPU is one of the main factors that limit the efficiency of geoscience numerical models in heterogeneous clusters. On the one hand, improving the data transfer bandwidth between CPU-GPU can improve the computational efficiency of the model in heterogeneous clusters. On the other hand, the optimization measures can be implemented to improve the data transfer efficiency between CPU-GPU. For example, (1) Asynchronous data transfer is used to reduce the communication latency between CPU and GPU. Computation and data transfer are performed simultaneously to hide communication overhead; (2) Currently, some advanced GPU architectures support a unified memory architecture,

so that the CPU and GPU can share the same memory space and avoid frequent data transfers. This reduces the overhead of data transfer and improves data transfer efficiency; (3) Cao et al. (2023) adopted communication optimization measures to reduce the communication frequency in one time integration step to one, but there is still the problem of high communication frequency in the whole simulation. In the future, we will consider porting other hotspots of CAMx model, or even the whole integral module except I/O, to GPU-like accelerators for increasing the proportion of code on the GPU and reduce the frequency of CPU-GPU communication. We have revised this part in **lines 412-428**, which are as follows:

**Lines 412-428:** *By comparing the kernel execution time and total running time of GPU-HADVPPM4HIP on the domestic accelerator, it can be seen that the data transfer efficiency between CPU and GPU is really inefficient, which seriously restricts the computational performance of numerical models in heterogeneous clusters. On the one hand, improving the data transfer bandwidth between CPU and GPU can improve the computational efficiency of the model in heterogeneous clusters. On the other hand, the optimization measures can be implemented to improve the data transfer efficiency between CPU and GPU. For example, (1) Asynchronous data transfer is used to reduce the communication latency between CPU and GPU. Computation and data transfer are performed simultaneously to hide communication overhead; (2) Currently, some advanced GPU architectures support a unified memory architecture, so that the CPU and GPU can share the same memory space and avoid frequent data transfers. This reduces the overhead of data transfer and improves data transfer efficiency; (3) Cao et al. (2023) adopted communication optimization measures to reduce the communication frequency in one time integration step to one, but there is still the problem of high communication frequency in the whole simulation. In the future, we will consider porting other hotspots of CAMx model, or even the whole integral module except I/O, to GPU-like accelerators for increasing the proportion of code on the GPU and reduce the frequency of CPU-GPU communication.*

**Line 339:**

until this point it was unclear to me what you were referring to with "super-large" clusters. Perhaps you could refer to this as heterogeneous supercomputers, or heterogeneous HPC systems.

**Response:** Thanks for the constructive suggestion. We have revised this part in **lines 441-442**, which are as follows:

**Lines 441-442:** *Generally, the heterogeneous HPC systems have thousands of compute nodes which are equipped with one or more GPUs on each compute node.*

**Line 362:**

which ones?

**Response:** Sorry for not being able to explain it clearly. We have revised this part in **lines 482-486**, which are as follows:

**Lines 482-486:** *As described in the Sect. 3.2, the horizontal advection module is accelerated by MPI and HIP technology, and the other modules, such as photolysis module, deposition module, chemical module, etc., which runs on the CPU are accelerated by MPI and OMP under the framework of the multi-level hybrid parallelism.*

**Line 378:**

Applications

**Response:** Sorry for this mistake. We have revised this part in **lines 500-503**, which are as follows:

**Lines 500-503:** *GPUs have become an essential part of providing processing power for high performance computing applications, especially in the field of geoscience numerical models, implementing super-large scale parallel computing of numerical models on GPUs has become one of the significant directions of its future development.*

**Line 380:**

Strikeout

**Response:** Sorry for this mistake. We have revised this part in **lines 503-507**, which are as follows:

**Lines 503-507:** *In this study, the ROCm HIP technology was implemented to port the GPU-HADVPPM from the NVIDIA GPUs to China's domestically GPU-like accelerators, and further introduced the multi-level hybrid parallelism scheme to improve the total computational performance of the CAMx-HIP model on the China's domestically heterogeneous cluster.*

**Line 374:**

I understand that these are OpenMP threads for the "other modules". What are the resources for the advection module? And for the air quality GPU kernels?

**Response:** Sorry for not being able to explain it clearly. As mentioned above, The original CAMx model supports message passing interface (MPI) parallel technology running on the general-purpose CPU. The simulation domain is divided into several sub-regions by MPI, and each CPU process is responsible for simulation of its sub-region, which includes advection module and other modules such as photolysis module, deposition module, chemical module, etc. In the previous studying, Cao et al. (2023) adopt a parallel architecture with an MPI and CUDA (MPI+CUDA) hybrid paradigm to configure one GPU accelerator for each CPU process. For the advection module, the simulation originally implemented by the CPU is handed over to the GPU. Other module computing tasks

continue to be completed on the CPU. In this study, in addition to the MPI+HIP hybrid parallel scheme to realize the advection module running on several domestic GPU-like accelerators, the OpenMP (OMP) hybrid parallel framework of CAMx model is further introduced to realize the MPI+OMP hybrid parallelism of other modules on CPU. For example, in a computing node, four CPU processes and four GPU-like accelerators are launched, and each CPU process spawns four threads. Then the advection module is simulated by 4 GPU-like accelerators, and the other modules are done by 4*4 threads spawned by CPU processes. We have modified this part in **lines 183-202**, which are as follows:

**Lines 183-202:** *As mentioned above, the original CAMx model supports message passing interface (MPI) parallel technology running on the general-purpose CPU. The simulation domain is divided into several sub-regions by MPI, and each CPU process is responsible for computation of its sub-region, which includes the computation tasks of advection module and other modules such as photolysis module, deposition module, chemical module, etc. In the previous studying, Cao et al. (2023) adopt a parallel architecture with an MPI and CUDA (MPI+CUDA) hybrid paradigm to configure one GPU accelerator for each CPU process. For the advection module, the simulation originally implemented by the CPU is handed over to the GPU. Other module computing tasks continue to be completed on the CPU.*

*In this study, when the CUDA C code of GPU-HADVPPM is converted to HIP C code, GPU-HADVPPM with an MPI and HIP (MPI+HIP) heterogeneous hybrid programming technology can also run on multiple domestic GPU-like accelerators. However, the number of GPU-like accelerators in a single compute node is usually much smaller than the number of CPU cores in the heterogeneous HPC systems. Therefore, in order to make full use of the remaining CPU computing resources, the OMP API of CAMx model is further introduced to realize the MPI+OMP hybrid parallelism of other modules on CPU. A schematic of the multi-level hybrid parallel framework is shown in Figure 2. For example, in a computing node, four CPU processes and four GPU-like accelerators are launched, and each CPU process spawns four threads. Then the advection module is simulated by 4 GPU-like accelerators, and the other modules are done by 4*4 threads spawned by CPU processes.*

**Lines 388-389:**

this is a very bold claim for which not enough evidence is provided

**Response:** Sorry for not being able to explain it clearly. In this paper, by comparing the simulation results of different versions of CAMx, it can be seen that GPU-HADVPPM4HIP has less deviation on domestic GPU-like accelerators. Because the simulation accuracy of geoscience numerical model is closely related to the model efficiency, and many model optimization works improve the computational performance by reducing the precision of the data, such as Váňa et al. (2017) changed some variables precision in the atmospheric model from double precision to single precision, which

increased the overall computational efficiency by 40%, and Wang et al. (2019) improved the computational efficiency of the gas-phase chemistry module in the air quality mode by 25%~28% by modifying the floating-point precision compile flag. Therefore, we speculate that this may be related to the manufacturing process of NVIDIA GPUs and domestic GPU-like accelerators, especially NIVIDA Tesla V100 series GPUs, which may use unknown optimizations to improve GPU performance efficiency by losing part of the accuracy. In this study, we mainly focus on numerical simulation, and the difference of hardware is not in the scope of our study. Of course, we also want to know the specific reasons for this, but we are not professional GPU research and development designers after all and do not know the underlying design logic of the hardware, so we can only present our experimental results in the air pollution model to you, and discuss with each other to jointly promote the application of GPU in the field of geoscience numerical models. We have modified this claim in **lines 508-512**, which are as follows:

**Lines 508-512:** *The consistency of model simulation results is a significant prerequisite for heterogeneous porting, although the experimental results show that the deviation between the CUDA version and the Fortran version of CAMx model, and the deviation between the HIP version and the Fortran version of CAMx model, are within the acceptable rang, the simulation difference between the HIP version of CAMx model and Fortran version of CAMx model is smaller.*

**Lines 387-388:**

difference between the HIP implementation and the CPU implementation?

**Response:** Sorry for not being able to explain it clearly. We have revised this part in **lines 508-512**, which are as follows:

**Lines 508-512:** *The consistency of model simulation results is a significant prerequisite for heterogeneous porting, although the experimental results show that the deviation between the CUDA version and the Fortran version of CAMx model, and the deviation between the HIP version and the Fortran version of CAMx model, are within the acceptable rang, the simulation difference between the HIP version of CAMx model and Fortran version of CAMx model is smaller.*

**Lines 386-387:**

difference between each other, or relative to something else (e.g., the CPU implementation?

**Response:** Sorry for not being able to explain it clearly. We have revised this part in **lines 386-388**, which are as follows:

**Lines 386-388:** *The consistency of model simulation results is a significant prerequisite for heterogeneous porting, although the experimental results show that the deviation between the CUDA version and the Fortran version of CAMx model, and the deviation between the HIP version*

*and the Fortran version of CAMx model, are within the acceptable rang, the simulation difference between the HIP version of CAMx model and Fortran version of CAMx model is smaller.*

**Line 390:**

relative to what?

**Response:** Sorry for not being able to explain it clearly. We have revised this part in **lines 512-515**, which are as follows:

**Lines 512-515:** *Moreover, the BJ, HN, and ZY test cases can achieve 8.5x, 11.5x, and 28.9x speedup, respectively, when the HADVPPM program is ported from the domestic CPU processor A to the domestic GPU-like accelerator A.*

**Line 391:**

Strikeout

**Response:** Sorry for this mistake. We have revised this part in **lines 515-518**, which are as follows:

**Lines 515-518:** *The experimental results of different cases show that the larger the computing scale, the more obvious the acceleration effect of the GPU-HADVPPM program, indicating that GPU is more suitable for super-large scale parallel computing, and provides technical support for accurate and fast simulation of ultra-high-resolution air quality at the meter level in the future.*

**Lines 391-393:**

this is to be expected, and is well known

**Response:** Sorry for not being able to explain it clearly. The experimental results of different cases show that the larger the computing scale, the more obvious the acceleration effect of the GPU-HADVPPM program, indicating that GPU is more suitable for super-large scale parallel computing, and provides technical support for accurate and fast simulation of ultra-high-resolution air quality at the meter level in the future. We have revised this part in **lines 515-518**, which are as follows:

**Lines 515-518:** *The experimental results of different cases show that the larger the computing scale, the more obvious the acceleration effect of the GPU-HADVPPM program, indicating that GPU is more suitable for super-large scale parallel computing, and provides technical support for accurate and fast simulation of ultra-high-resolution air quality at the meter level in the future.*

**Line 395:**

as shown by the fact that the

**Response:** Sorry for this mistake. We have revised this part in **lines 518-523**, which are as follows:

**Lines 518-523:** *The data transfer bandwidth between CPU and GPU is one of the most important factors affecting the computational efficiency of numerical model in heterogeneous clusters, as shown by the fact that the elapsed time of GPU-HADVPPM program on GPU only accounts for 7.3% and 23.8% when considering the data transfer time between CPU and GPU on the the "Songshan" supercomputer and "Taiyuan" computing platform.*

**Lines 396-397:**

do you mean that only 7.3 and 23.8 of the runtime is spent on computations? the rest is data transfers?

**Response:** Sorry for not being able to explain it clearly. Data transfer between CPU-GPU is indeed one of the main factors that limit the efficiency of geoscience numerical models in heterogeneous clusters, and we will plan to carry out CPU-GPU data transfer optimization in the following three aspects in the future. (1) Asynchronous data transfer is used to reduce the communication latency between CPU and GPU. Computation and data transfer are performed simultaneously to hide communication overhead; (2) Currently, some advanced GPU architectures support a unified memory architecture, so that the CPU and GPU can share the same memory space and avoid frequent data transfers. This reduces the overhead of data transfer and improves data transfer efficiency; (3) Cao et al. (2023) adopted communication optimization measures to reduce the communication frequency in one time integration step to one, but there is still the problem of high communication frequency in the whole simulation. In the future, we will consider porting other hotspots of CAMx model, or even the whole integral module except I/O, to GPU-like accelerators for increasing the proportion of code on the GPU and reduce the frequency of CPU-GPU communication. We have revised this part in **lines 412-428**, which are as follows:

**Lines 412-428:** *By comparing the kernel execution time and total running time of GPU-HADVPPM4HIP on the domestic accelerator, it can be seen that the data transfer efficiency between CPU and GPU is really inefficient, which seriously restricts the computational performance of numerical models in heterogeneous clusters. On the one hand, improving the data transfer bandwidth between CPU and GPU can improve the computational efficiency of the model in heterogeneous clusters. On the other hand, the optimization measures can be implemented to improve the data transfer efficiency between CPU and GPU. For example, (1) Asynchronous data transfer is used to reduce the communication latency between CPU and GPU. Computation and data transfer are performed simultaneously to hide communication overhead; (2) Currently, some advanced GPU architectures support a unified memory architecture, so that the CPU and GPU can share the same memory space and avoid frequent data transfers. This reduces the overhead of data transfer and improves data transfer efficiency; (3) Cao et al. (2023) adopted communication optimization measures to reduce the communication frequency in one time integration step to one, but there is still the problem of high communication frequency in the whole simulation. In the future,*

*we will consider porting other hotspots of CAMx model, or even the whole integral module except I/O, to GPU-like accelerators for increasing the proportion of code on the GPU and reduce the frequency of CPU-GPU communication.*

**Line 405:**

could be

**Response:** Sorry for this mistake. We have revised this part in **lines 530-532**, which are as follows:

**Lines 530-532:** *Finally, the data type of some variables could be changed from double precision to single precision, and the mixing-precision method is used to further improve the CAMx-HIP computing performance.*

---

## Author Response (AR2)

**Response to Reviewers' comments**

We thank the two reviewers for their thoughtful and constructive comments that helped us substantially improve the manuscript. We have revised the manuscript accordingly. Listed below is our blue point-to-point response to each comment the reviewers offered.

**Response to Reviewer #1**

The authors have adequately addressed all comments and implemented a number of suggestions that have substantially improved the manuscript and presentation of results. I thus recommend that the manuscript is accepted for publication, subject to the call of the professional English language services for editortial review.

**Response:** Thanks a million for your precious time and your suggestion. To improve the English language of the manuscript, we have carefully corrected the grammatical errors in the paper. On the other hand, we will agree to apply the typesetting and language copy-editing service provided by Copernicus Publications during the production (https://www.geoscientific-model-development.net/policies/proofreading_guidelines.html). Moreover, two reviewers' point-to-point responses are listed below.

**Response to Reviewer #2**

The authors have addressed most of the points I raised.

Nevertheless, there are still two major not-fully-resolved points.

**Response:** We appreciate the editor for reviewing our manuscript and the valuable suggestions, which we will address point by point in the following.

1- On the point of accuracy vs performance on NVIDIA hardware:

My main point of criticism was, in a nutshell, that the evidence provided (Figure 5) is not specific, thorough or deep enough to support the claim that the different range of errors in the species can be attributed to the hardware. The authors have included some comments on the tradeoffs between accuracy and computational performance, by invoking mixed precision techniques (which by their own account, they are not using), and now "speculate" that the differences compared to CPU

computations can be explained by manufacturing processes resulting in unknown optimisation on GPU performance while renouncing accuracy. This is only a very minor toning-down of the previous claim. The authors also claim not to have sufficient expertise and knowledge to attribute this.

This is precisely the point. The claims are in my opinion still too bold. The authors seem to be sure that nothing else except hardware differences can explain the differences in the relative errors, which is a big leaf of faith from Figure 5. I would argue there are many things that can go wrong, and because there are basically two different codes at play, it is likely. At the very least, it cannot be simply assumed that nothing else except hardware is to blame.

An example: the authors point out that only the advection module is running on GPUs, and other modules on CPUs (including the chemistry). They also point out that for PNH4, PNO3 and PSO4 the initial concentrations are very low and they are generated via complex chemical reactions. I translate this as follows: the reactive part of the equations is very important. Presumably, reactions are computed in the chemistry module, which is always on the CPUs. Then, why would the GPUs be mostly responsible for these errors which are far away from machine precision? I find it hard to attribute such large errors to porting an advection solver to GPUs (in fact, my experience is quite the opposite, accuracy is very much comparable).

I unfortunately cannot take my own argument much further without making assumptions about the solvers and implementations. For example, if the chemical reactions lead to iterative linearisations and linear solvers, some convergence tolerances can play a role in there, which may be orders of magnitude above machine precision. This would not necessarily explain the difference between the Nvidia and the Chinese accelerators, but the errors could come from the different behaviours on the CPU.

I concede that all of this is speculation on my side, but I write it to illustrate the variety of thoughts one can have before attributing these differences to hardware. I do not intend to claim that it is NOT attributable to hardware, but proving that would require to define other types of tests, which offer much more control.

**Response:** Thanks for the constructive comment. To further detail the differences in the simulation results, we supplement the offline experimental results of the advection module on the NVIDIA K40m cluster and the Songshan supercomputer. First, we construct the Fortran programs

to provide consistent input data for the advection module written in CUDA C code and HIP C code on NVIDIA Tesla K40m GPU and domestic GPU-like accelerator, respectively. The accuracy of the input data is kept at 12 decimal places. Then, the advection module outputs and prints the computing results after completing one integration operation on different accelerators. Finally, the results of the various accelerators were compared with those of the Fortran code on the Intel Xeon E5-2682 v4 CPU processor. The specific results are shown in the Fig.5 below. The difference in the computing results of the advection module written in HIP C code on the domestic GPU-like accelerator is smaller than that of the CUDA C code on the NVIDIA Tesla K40m GPU. The mean relative errors (REs) and AEs of the computing results on the NVIDIA Tesla K40m GPU are $1.3 \times 10^{-5}\%$ and $7.1 \times 10^{-9}$, respectively, while on the domestic GPU-like accelerator, the mean REs and AEs of the results are $5.4 \times 10^{-6}\%$ and $2.6 \times 10^{-9}$, respectively. The above codes related to offline tests are available online via ZENODO (https://zenodo.org/doi/10.5281/zenodo.10158214).

In the air quality model, the initial concentration of secondary fine particulate matter such as $PSO_4$, $PNO_3$, and $PNH_4$ is very low and is mainly generated by complex chemical reactions. The integration process of the advection module is ported from the CPU processor to the GPU accelerator, which will lead to minor differences in the results due to different hardware. The low initial concentration of secondary fine particulate matter is sensitive to these minor differences, which may eventually lead to a higher difference in the simulation results of secondary particulate matter than other species. We have modified this part in **lines 277-313**, which are as follows:

**Lines 277-313:**

*To further detail the differences in the simulation results, we supplement the offline experimental results of the advection module on the NVIDIA K40m cluster and the Songshan supercomputer. First, we construct the Fortran programs to provide consistent input data for the advection module written in CUDA C code and HIP C code on NVIDIA Tesla K40m GPU and domestic GPU-like accelerator, respectively. The accuracy of the input data is kept at 12 decimal places. Then, the advection module outputs and prints the computing results after completing one integration operation on different accelerators. Finally, the results of the various accelerators were compared with those of the Fortran code on the Intel Xeon E5-2682 v4 CPU processor. The specific results are shown in the Fig.5. The difference in the computing results of the advection*

*module written in HIP C code on the domestic GPU-like accelerator is smaller than that of the CUDA C code on the NVIDIA Tesla K40m GPU. The mean relative errors (REs) and AEs of the computing results on the NVIDIA Tesla K40m GPU are $1.3 \times 10^{-5}\%$ and $7.1 \times 10^{-9}$, respectively, while on the domestic GPU-like accelerator, the mean REs and AEs of the results are $5.4 \times 10^{-6}\%$ and $2.6 \times 10^{-9}$, respectively.*

*Fig.6 further presents the boxplot of the REs in all grid boxes for the $PSO_4$, $PNO_3$, $PNH_4$, $O_3$, CO, and $NO_2$ during the 48-hour simulation under the BJ case. Statistically, the REs between the CUDA version on the NVIDIA K40m cluster and Fortran version on the Intel E5-2682 v4 CPU for the above six species are in the range of $\pm 0.006\%$, $\pm 0.01\%$, $\pm 0.008\%$, $\pm 0.002\%$, $\pm 0.002\%$, and $\pm 0.002\%$. In terms of REs between the HIP version on the "Songshan" supercomputer and the Fortran version on the Intel E5-2682 v4 CPU, the values are much smaller than REs between CUDA and Fortran versions which fall into the range of $\pm 0.0005\%$, $\pm 0.004\%$, $\pm 0.004\%$, $\pm 0.00006\%$, $\pm 0.00004\%$, and $\pm 0.00008\%$, respectively. In the air quality model, the initial concentration of secondary fine particulate matter such as $PSO_4$, $PNO_3$, and $PNH_4$ is very low and is mainly generated by complex chemical reactions. The integration process of the advection module is ported from the CPU processor to the GPU accelerator, which will lead to minor differences in the results due to different hardware. The low initial concentration of secondary fine particulate matter is sensitive to these minor differences, which may eventually lead to a higher difference in the simulation results of secondary particulate matter than other species.*

[Figure]

*Figure 5.* *The boxplots of REs and AEs between the Fortran code on Intel Xeon E5-2682 v4 CPU and CUDA C code on NVIDIA Tesla K40m GPU, and between HIP C code on domestic GPU-like accelerator, respectively, in the case of offline testing.*

[Figure]

*Figure 6.* *The REs distribution in all grid boxes for the $PSO_4$, $PNO_3$, $PNH_4$, $O_3$, CO, and $NO_2$ under the BJ case. The red boxplot represents the REs between the CUDA version on the NVIDIA K40m cluster and the Fortran version on the Intel E5-2682 v4 CPU, and the blue boxplot represents the REs between the HIP version on the*

2- this point is far simpler. It remains still unclear in section 4.3.1 exactly how many computational resources are used (e.g., how many cores for CPU computations, it seems only single GPUs). It might be good to simply state this very explicitly, very much how it is done in section 4.3.2. Nevertheless, in both sections explicitly stating how each module maps to which and how much hardware would be very much appreciated.

**Response:** Sorry for not being able to explain it clearly. We only used one CPU core when testing the computation performance of the advection module on the different GPU accelerator in section 4.3.1. For the MPI and HIP heterogeneous hybrid parallelization technology as described in section 4.3.2, we first use the ROCm-HIP library function hipGetDeviceCount to obtain the number of GPU accelerators configured for each compute node, and then the total number of accelerators to be launched and the ID number of accelerator cards in each node were determined according to the MPI process ID number and the remainder function in standard C language. Finally, the hipSetDevice library function in ROCm-HIP is used to configure an accelerator for each CPU core. During the simulation of the CAMx model, the computing of the emission, advection, dry deposition, diffusion, wet deposition, photolysis process and chemical process will be completed sequentially. In heterogeneous computing platforms, except for the advection process, the CPU processor completes the simulation of the rest of the processes, and the advection process is completed on the GPU accelerator. For example, using MPI and HIP hybrid parallel technology to launch four CPU processes and four GPU accelerators simultaneously, the advection process is completed on four GPU, and the other processes are still completed on four CPU processes. We have modified this part in **lines 182-192, lines 363-366 and lines 455-464**, which are as follows:

**Linse 182-192:**

*As mentioned, the original CAMx model supports message passing interface (MPI) parallel technology running on the general-purpose CPU. The simulation domain is divided into several sub-regions by MPI, and each CPU process is responsible for the computation of its sub-region. To expand the heterogeneous parallel scale of the CAMx model on the Songshan supercomputer, a hybrid parallel architecture with an MPI and HIP was adopted to make full use of GPU*

*computing resources. Firstly, we use the ROCm-HIP library function hipGetDeviceCount to obtain the number of GPU accelerators configured for each compute node. Then, the total number of accelerators to be launched and the ID number of accelerator cards in each node were determined according to the MPI process ID number and the remainder function in standard C language. Finally, the hipSetDevice library function in ROCm-HIP is used to configure an accelerator for each CPU core.*

**Lines 363-366:**

*When comparing the speedup on different GPU accelerators, the elapsed time of the advection module launched one CPU process (P1) on the domestic CPU processor A is taken as the benchmark; that is, the speedup is 1.0x. The runtime of the advection module on Intel CPU processor and different GPU accelerators is compared with the baseline to obtain the speedup.*

**Lines 455-464:**

*Generally, heterogeneous HPC systems have thousands of compute nodes equipped with one or more GPUs on each compute node. To fully use multiple GPUs, the hybrid parallelism with an MPI and HIP paradigm was used to implement the HIP version of GPU-HADVPPM run on multiple domestic GPU-like accelerators. During the simulation of the CAMx model, the emission, advection, dry deposition, diffusion, wet deposition, photolysis process, and chemical process will be computed sequentially. In heterogeneous computing platforms, except for the advection process, the CPU processor completes the simulation of the rest of the processes, and the advection process is completed on the GPU accelerator. For example, using MPI and HIP hybrid parallel technology to launch four CPU processes and four GPU accelerators simultaneously, the advection process is completed on four GPUs, and the other processes are still completed on four CPU processes.*